# LITTLE BY LITTLE: CONTINUAL LEARNING VIA SELF-ACTIVATED SPARSE MIXTURE-OF-RANK ADAPTIVE LEARNING

## ABSTRACT

Continual learning (CL) with large pre-trained models is challenged by catastrophic forgetting and task interference. Existing LoRA-based Mixture-of-Experts (MoE) approaches mitigate forgetting by assigning and freezing task-specific adapters, but suffer from interference, redundancy, and ambiguous routing due to coarse adapter-level selection. However, this design introduces three key challenges: 1) *Interference*: Activating full LoRA experts per input leads to subspace interference and prevents selective reuse of useful components across tasks. 2) *Redundancy*: Newly added experts often duplicate or contradict existing knowledge due to unnecessary activation of unrelated ranks and insufficient reuse of relevant ones. 3) *Ambiguity*: Overlapping features across tasks confuse the router, resulting in unstable expert assignments. As more experts accumulate, earlier task routing degrades, accelerating forgetting. We propose *MoRA*, a **M**ixture-**o**f-**R**ank **A**daptive learning approaches with self-activated and sparse rank activation for CL. Unlike mixing multiple low-rank matrices, MoRA decomposes each rank-$r$ update into $r$ rank-one components, each treated as an independent expert, enabling fine-grained rank-one expert utilization while mitigating interference and redundancy. To avoid ambiguous routing, we propose that each rank-one expert can infer its own relevance via intermediate activations. Coupled with our proposed rank pruning and activation budgets, MoRA adaptively selects a sparse mixture of ranks per input. We validate MoRA on continual learning benchmarks using CLIP and language models, analyzing both in-domain learning and out-of-domain forgetting/generalization during fine-tuning. MoRA shows significant effectiveness on enhancing CL with PTMs, and improving generalization while mitigating forgetting.

*"Little by little, we gave you everything you ever dreamed of ..."* — *"Little by little"*, Oasis.

## 1 INTRODUCTION

Continual learning (CL) (Hadsell et al., 2020; De Lange et al., 2021; Ding et al., 2022) aims to enable models to incrementally and efficiently acquire new knowledge from a stream of tasks, without catastrophic forgetting (Nguyen et al., 2019; McCloskey & Cohen, 1989) or the need for repeated fine-tuning on all previously seen data (Wang et al., 2022f; 2024). The emergence of large pre-trained models (PTMs), including Vision Transformer (ViT) for vision (Dosovitskiy et al., 2020), Language Model (LM) (Grattafiori et al., 2024; Raffel et al., 2020), and CLIP for vision–language embeddings (Radford et al., 2021), has spurred a wave exploring CL with PTMs (Xu et al., 2024; Wang et al., 2023a). Despite the success of PTMs, CL on them across sequential tasks still faces key challenges, including preventing catastrophic forgetting and preserving, or even enhancing, the pre-trained capabilities to generalize to unseen or future cases (Qiao & Mahdavi, 2024).

A common way to integrate new knowledge into PTMs is full fine-tuning (Zheng et al., 2023; Garg et al., 2023) or PEFT with a small set of learnable parameters (Yang et al., 2024b), such as LoRA (Hu et al., 2021) (Fig. 1(a)). However, continually updating the same parameters often leads to interference and forgetting, even with a single new task (Biderman et al., 2024). Leveraging LoRA's regularization and flexibility, many CL methods adopt LoRA or its variants with additional strategies to mitigate forgetting (Yu et al., 2024; Wang et al., 2023a; Qiao & Mahdavi, 2024).

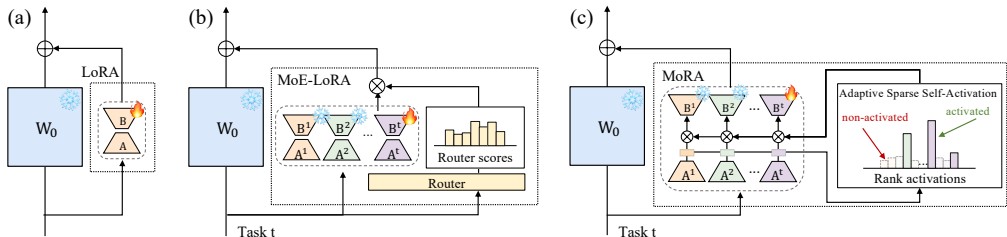

Figure 1: Conceptional illustration of CL with (a) LoRA, (b) MoE-LoRA, and (c) MoRA (Ours).

Inspired by mixture-of-experts (MoE) architectures (Shazeer et al., 2017; Lepikhin et al., 2020; Fedus et al., 2022; Dai et al., 2024) and the modular nature of LoRA adapters, MoE frameworks using LoRA adapters as experts have been studied (Dou et al., 2023; Wu et al., 2024b) and have been widely adopted in CL (Yu et al., 2024; Wang et al., 2024; Yang et al., 2024a; Chen et al., 2024; Li et al., 2025) (Fig. 1(b)). These works either predefine an MoE with LoRA for CL (Yang et al., 2024a), or incrementally add experts (Wang et al., 2024) or task-specific routers (Yu et al., 2024), assuming MoE benefits CL by isolating task interference. Such methods (Wang et al., 2024; Qiao & Mahdavi, 2024; Rusu et al., 2016; Yu et al., 2024) freeze old components and add new ones (e.g., experts or routers) to reduce forgetting. Despite design differences, we collectively refer a plain and general design with LoRA-based MoE as MoE-LoRA. In MoE-LoRA models, each expert is a LoRA adapter with multiple ranks (subspaces), and the router selects among experts with each LoRA adapter as a unit. Despite their success, we argue that the coarse granularity of experts and routing introduces three key challenges that limit effective utilization:

**(1) Interference across experts and limited knowledge reuse.** A coarse-grained multi-rank LoRA expert is packed with a rich set of knowledge that must be fully activated or not at all. For each sample (*i.e.*, token), any two experts may hold both complementary and conflicting knowledge, creating a dilemma: selecting both causes interference, while only selecting either one prevents reuse.

**(2) Learning redundancy and conflicts.** As a result of (1), (newly added) experts often have to carry conflicting or redundant information, either from activating other experts with both useful and unnecessary content or from insufficient reuse of existing ones. This coarse selection exacerbates interference and forgetting during inference.

**(3) Routing ambiguity and forgetting.** With coarse-grained experts and routing, redundancy and conflicts hinder precise input–module assignment, leading to suboptimal mixtures (Lepikhin et al., 2020; Fedus et al., 2022; Dai et al., 2024). As more experts accumulate in CL, router assignments for earlier tasks grow unreliable, increasing forgetting.

The three challenges are intertwined causes and effects of coarse-grained experts and routing, limiting the potential of promising MoE-LoRA design.

We propose **MoRA**, a **M**ixture-**o**f-**R**ank **A**daptive learning approaches with self-activated and sparsely selected ranks across the LoRAs added at different tasks. To address the limitations of coarse-grained MoE-LoRA methods, MoRA operates at a fine-grained level by treating each rank-one subspace within a LoRA adapter as an individual expert (or treating each rank-one LoRA as an expert). Expert selection and routing occur at the rank level, avoiding interference from whole LoRA blocks. This decomposition promotes specialization on narrower data regions, reduces redundancy, and enables effective knowledge reuse across tasks. However, significantly increasing the number of experts poses a challenge for traditional external routers, which often suffer from optimization difficulties and ambiguity. To resolve this, we introduce a self-activation mechanism to select a sparse set of ranks without relying on a separately trained router. By defining each rank-one component as an expert, we avoid a separate router for expert assignment: the rank-one $\mathbf{A}$ in LoRA can act as a weighting router for its $\mathbf{B}$. Each rank evaluates its relevance to the input and determines its contribution, grounded in the view of low-rank updates as linear key–value memory. This design unifies rank-one LoRA with fine-grained MoE while preserving the original characteristics, saving parameters (for additional router), reducing routing confusion, and further mitigating forgetting (Fig. 1(c)). Sparsity is enforced via temperature-scaled softmax together with `top-k` masking and rank pruning under a rank activation budget. As a complement, test-time thresholding further improves adaptability across cases. Leveraging the connection between PTMs and low-dimensional manifolds

(Li et al., 2018; Aghajanyan et al., 2020), MoRA enables precise, input-specific adaptation through selective subspace updates. With fine-grained rank-level experts and sparse activation, MoRA enables continual learning by reusing old ranks and selectively activating a few new ones per sample, allowing efficient "little-by-little" adaptation with minimal redundancy. This sample-specific activation at inference also preserves pre-trained knowledge and improves generalization to unseen tasks.

We validate MoRA on CL with both vision–language CLIP and LMs, analyzing its impact on pre-trained knowledge during low-rank fine-tuning. MoRA delivers strong downstream performance with far fewer active parameters, greatly reduces forgetting of prior tasks and pre-trained knowledge, and even improves generalization to unseen domains.

## 2 RELATED WORK

**Continual learning** enables sequential knowledge acquisition without forgetting. Experience replay (ER) methods (Luo et al., 2023; Aljundi et al., 2019b; Chaudhry et al., 2018a; Liu et al., 2020; Chaudhry et al., 2018b; Yan et al., 2022; 2021) interleave past examples with new data. Parameter regularization (Kirkpatrick et al., 2017; Aljundi et al., 2018; Zenke et al., 2017; Aljundi et al., 2019a; Jha et al., 2023) penalizes updates to critical weights. Dynamic networks (Zhou et al., 2024; Wang et al., 2022a;b; Zhou et al., 2022; Wang et al., 2024; McDonnell et al., 2024; Liang & Li, 2024; Wang et al., 2022f;e; Smith et al., 2023; Wang et al., 2022c; 2024) allocate new capacity on the fly and preserve dedicated pathways for prior tasks.

**Continual learning of PTMs.** For CL on vision–language CLIP model (Jha et al., 2024; Zhang et al., 2024b; Garg et al., 2023), methods like ZSCL (Zheng et al., 2023) retain zero-shot performance during adaptation, and follow-up work (Yu et al., 2024; Tang et al., 2025; Xu et al., 2024; Lu et al., 2024; Wu et al., 2025) continually fine-tunes while leveraging frozen pre-trained predictions. The X-TAIL benchmark (Xu et al., 2024) further challenges models by mixing domain labels at test time. In language models (LMs) (de Masson D'Autume et al., 2019; Qin & Joty, 2021; Razdaibiedina et al., 2023; Wang et al., 2023a; Qiao & Mahdavi, 2024), continual learning uses capacity expansion or task-specific submodules to reduce interference.

**Low-rank adaptation** (LoRA) (Hu et al., 2021) is widely used for parameter-efficient fine-tuning of large pre-trained models. Building on this foundation, recent methods reformulate LoRA's updates via SVD-based initialization and dynamic rank scheduling (Zhang et al., 2024a; Meng et al., 2024; Ding et al., 2023; Zhang et al., 2023; Liu et al., 2024; Wu et al., 2024a), showing that task adaptation primarily lies in a compact subspace. In this work, we offer a complementary perspective by viewing both pre-trained weight matrix and its low-rank updates as a linear associative memory (Li et al., 2018; Aghajanyan et al., 2020; Kohonen, 1972; Anderson, 1972). Under this lens, a rank-$r$ update corresponds to writing $r$ new key–value entries into the pre-trained memory matrix, with each rank-one component acting as an independent memory slot.

**Mixture-of-experts (MoE) in LoRA fine-tuning.** MoE increases model capacity by replacing the Transformer's feed-forward layer with expert subnetworks, routing each input to a small subset and using load-balancing losses to even out usage (Shazeer et al., 2017; Lepikhin et al., 2020; Fedus et al., 2022; Dai et al., 2024). This paradigm has been adapted for standard fine-tuning (Chen et al., 2023; Li et al., 2024; Dou et al., 2023) and CL (Yu et al., 2024; Wang et al., 2024; Yang et al., 2024a; Chen et al., 2024) with LoRA, treating each low-rank adapter as an expert: new adapters are trained and frozen per task to prevent forgetting. In contrast, we decompose each rank-$r$ update into $r$ rank-1 components and compute an input-dependent mixture over these fine-grained experts, greatly enhancing both expert specialization and mixture diversity.

## 3 METHODS

### 3.1 PRELIMINARIES

**Continual learning.** In CL, a model sequentially learns $T$ tasks. For task $t \in 1, \ldots, T$, let $\mathcal{D}^t = \{(\mathbf{x}_i^t, y_i^t)\}_{i=1}^{N^t}$, where $\mathbf{x}_i^t \in \mathbb{R}^{n \times d}$, $y_i^t \in \mathcal{C}^t$, and $N^t$ is the number of examples. In the memory-free setting, the model may access only $\mathcal{D}^t$ and cannot access data from any $\mathcal{D}^u$ with $u < t$.

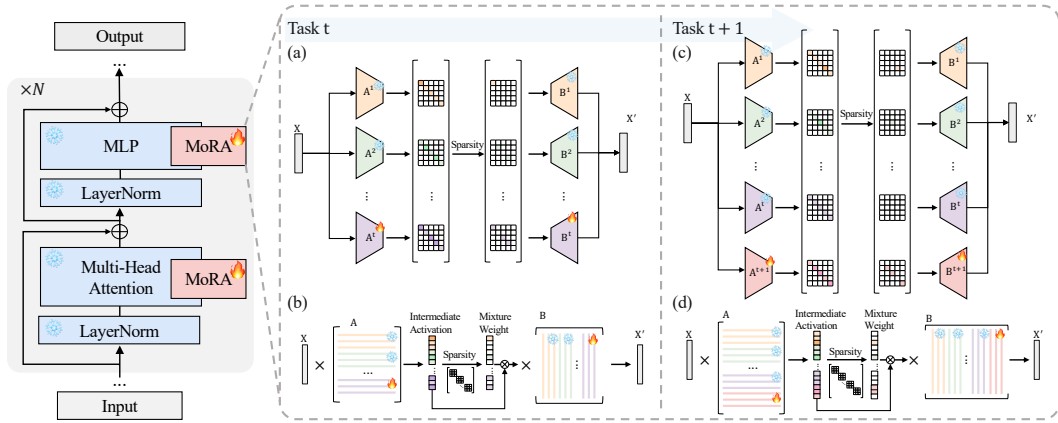

Figure 2: Overview of MoRA. For each new task, we freeze the ranks learned on previous tasks and introduce $r$ new ranks of updates. Our sparse self-activated mixture-of-ranks framework jointly considers all old and new ranks, adaptively inferring a sparse mixture weight for each rank. Panels (a,c) illustrate MoRA conceptually and (b,d) detail its computation for tasks $t$ and $t + 1$, respectively.

**Low-rank adaptation.** LoRA (Hu et al., 2021) parameterizes a low-rank update to a pre-trained weight matrix $\mathbf{W}_0 \in \mathbb{R}^{d_{out} \times d_{in}}$ by introducing two factors $\mathbf{B} \in \mathbb{R}^{d_{out} \times r}$ and $\mathbf{A} \in \mathbb{R}^{r \times d_{in}}$, such that $\Delta \mathbf{W} = \mathbf{B}\mathbf{A}$, where $r \ll \min(d_{in}, d_{out})$. The updated weight matrix is then defined as:

$$\mathbf{W} = \mathbf{W}_0 + \Delta \mathbf{W} = \mathbf{W}_0 + \mathbf{B}\mathbf{A}. \tag{1}$$

In this formulation, the original weights $\mathbf{W}_0$ remain fixed, and only $\mathbf{B}$ and $\mathbf{A}$ are trained, reducing the number of trainable parameters from $d_{in}d_{out}$ to $r(d_{in} + d_{out})$.

**Mixture-of-Experts LoRA.** Building on the Mixture-of-Experts (MoE) paradigm, a common generic framework of MoE-LoRA (Yu et al., 2024; Wang et al., 2024) treats each LoRA as an independent expert. Suppose after $T$ tasks, we have $T$ LoRA experts $\{(\mathbf{A}^1, \mathbf{B}^1), \ldots, (\mathbf{A}^T, \mathbf{B}^T)\}$. For an input token $x \in \mathbb{R}^{d_{in}}$, the overall LoRA update in this framework is given by $\Delta \mathbf{W} = \sum_{i=1}^{T} R(x)_i \mathbf{B}^i \mathbf{A}^i$, where the mixture weight $R(x) \in \mathbb{R}^T$ is produced by a learned router $R(\cdot) = \text{softmax}(x\mathbf{W}_r)$ and $\mathbf{W}_r \in \mathbb{R}^{d_{in} \times T}$ contains the router's trainable parameters. Each LoRA's contribution is weighted by the learnable router, enabling the model to dynamically select and combine the most relevant low-rank updates for each token. In practice, a Top-$k$ masking is typically applied to mixture weights to enforce sparsity, activating only the $k$ most relevant experts and controlling computational overhead.

## 3.2 MODEL WEIGHTS AND UPDATES AS LINEAR ASSOCIATIVE MEMORY MODEL

**Weight matrix as linear associative memory.** Large pre-trained models (PTMs) have been shown to inhabit a low "intrinsic-dimension" manifold compared to their full parameter count (Li et al., 2018; Aghajanyan et al., 2020). Any weight matrix of a pre-trained model $\mathbf{W}_0 \in \mathbb{R}^{d_{out} \times d_{in}}$ can be treated as a linear associative memory model with a key-value structure (Kohonen, 1972; Anderson, 1972; Bau et al., 2020; Meng et al., 2022). Concretely, for any matrix $\mathbf{W} \in \mathbb{R}^{d_{out} \times d_{in}}$ and input $\mathbf{x} \in \mathbb{R}^{d_{in}}$, the product $\mathbf{W}\mathbf{x}$ implements a content-addressable memory read over $m$ stored key–value pairs $\{(\mathbf{K}_{i,:}, \mathbf{V}_{:,i})\}_{i=1}^{m}$, with $m$ potentially large and varied across each weight matrix. Here, $\mathbf{K} = \begin{bmatrix} \mathbf{K}_{1,:}^{\top} & \cdots & \mathbf{K}_{m,:}^{\top} \end{bmatrix}^{\top} \in \mathbb{R}^{m \times d_{in}}$ and $\mathbf{V} = [\mathbf{V}_{:,1} \quad \cdots \quad \mathbf{V}_{:,m}] \in \mathbb{R}^{d_{out} \times m}$ with $\mathbf{K}_{i,:} \in \mathbb{R}^{1 \times d_{in}}$ is the $i$-th row of $\mathbf{K}$ and $\mathbf{V}_{:,i} \in \mathbb{R}^{d_{out} \times 1}$ is the $i$-th column of $\mathbf{V}$. Given any input $\mathbf{x}$, the output $\mathbf{y} \in \mathbb{R}^{d_{out}}$ is computed as

$$\mathbf{y} = \mathbf{W}\mathbf{x} \approx \sum_{i=1}^{m} \mathbf{V}_{:,i}(\mathbf{K}_{i,:}\mathbf{x}). \tag{2}$$

In this formulation, the forward pass through the weight matrix with the input decomposes into two steps: (1) computing response scores or assignment weights via the key vectors $\mathbf{K}_{i,:}$, and (2) aggregating the corresponding value vectors $\mathbf{V}_{:,i}$ according to those weights.

**Parameter update as modifications to the linear associative memory.** Building on the associative-memory view of weights, we can likewise interpret weight updates during training as key–value

insertions and modification into the existing memory. During training, for an arbitrary update $\Delta\mathbf{W}$ on a given $\mathbf{W}_0$ as the starting point, the weight update process can be represented as

$$\mathbf{W}_{\text{new}} = \mathbf{W}_0 + \Delta\mathbf{W}, \tag{3}$$

where $\mathbf{W}_0$ can be the pre-trained model weights. We can also formulate the weight updates $\Delta\mathbf{W}$ with the key-value memory model $\{\Delta\mathbf{K}, \Delta\mathbf{V}\}$ as an update to the original memory model. With the operation on an input $\mathbf{x}$ via

$$\Delta\mathbf{W}\mathbf{x} = \Delta\mathbf{V}\Delta\mathbf{K}\mathbf{x}, \tag{4}$$

the updates on weights can be seen as a retrievable key-value system for the updating of $\mathbf{W}\mathbf{x}$.

**LoRA as low-rank key–value memory updates.** LoRA (Hu et al., 2021) performs parameter-efficient fine-tuning by adding a rank-$r$ update $\Delta\mathbf{W} = \mathbf{B}\mathbf{A}$ to the pre-trained weight matrix $\mathbf{W}_0$, where $\mathbf{B} \in \mathbb{R}^{d_{\text{out}} \times r}$ and $\mathbf{A} \in \mathbb{R}^{r \times d_{in}}$. With the model update based on LoRA, given an input $\mathbf{x}$, the computation of updated weight becomes

$$\mathbf{y}_{\text{new}} = \mathbf{W}_{\text{new}}\mathbf{x} = \mathbf{W}_0\mathbf{x} + \Delta\mathbf{W}\mathbf{x} = \mathbf{W}_0\mathbf{x} + \mathbf{B}\mathbf{A}\mathbf{x}. \tag{5}$$

Decomposing the low-rank matrices into a collection of its $r$ rank-one components gives

$$\Delta\mathbf{W}\mathbf{x} = \mathbf{B}\mathbf{A}\mathbf{x} = \sum\nolimits_{i=1}^{r} \mathbf{B}_{:,i}(\mathbf{A}_{i,:}\mathbf{x}), \tag{6}$$

with $\mathbf{A} = \begin{bmatrix} \mathbf{A}_{1,:}^{\top} & \cdots & \mathbf{A}_{r,:}^{\top} \end{bmatrix}^{\top} \in \mathbb{R}^{r \times d_{\text{in}}}$ where $\mathbf{A}_{i,:}$ is the $i$-th row of $\mathbf{A}$, and $\mathbf{B} = \begin{bmatrix} \mathbf{B}_{:,1} & \cdots & \mathbf{B}_{:,r} \end{bmatrix} \in \mathbb{R}^{d_{\text{out}} \times r}$ where $\mathbf{B}_{:,i}$ is the $i$-th column of $\mathbf{B}$.

Considering the connection between the formulation of LoRA Eq. (1) and associative-memory view (Anderson, 1972; Meng et al., 2022) of weight update Eq. (4), we can intuitively interpret LoRA as obtaining a small key–value store in a low-rank subspace, where the rows of $\mathbf{A}$ serve as *keys* and the columns of $\mathbf{B}$ serve as their corresponding *values*. After training the LoRA module on a base weight matrix $\mathbf{W}_0$, the modified forward pass in Eq. (5) with Eq. (6) can be seen as: first, the original pre-trained computation $\mathbf{W}_0\,\mathbf{x}$ (Eq. (2)) is performed; then, each learned key $\mathbf{A}_{i,:}$ measures its relevance to the input $\mathbf{x}$ via the product $\mathbf{A}_{i,:}\mathbf{x}$, and each learned value $\mathbf{B}_{:,i}$ is scaled by this relevance score and aggregated, in a same way as shown in Eq. (2). In this way, LoRA effectively inserts or updates a small set of key–value memories that augment the pre-trained model's capabilities.

### 3.3 Mixture of Ranks Adaptation

In the key–value associative-memory view of LoRA (Eq. (5) and Eq. (6)), a standard LoRA or its variants (Hu et al., 2021; Meng et al., 2024; Zhang et al., 2024a) apply a dense update: every rank's subspace contributes to $\Delta\mathbf{W}\,\mathbf{x}$ for any input $\mathbf{x}$. Even in MoE-LoRA, where each LoRA module is treated as a separate expert, all key–value pairs within a LoRA module are typically used densely as a whole (either fully activated or not), which creates redundancy across experts and underutilizes the expressive potential of the key vectors $\mathbf{A}$ to indicate input relevance.

Rather than by-default activating all key–value memories for every input, many of which may encode patterns irrelevant to the current token, we introduce a gated, sparse mixture-of-ranks update that selectively activates only the most relevant rank-one key–value pairs. Concretely, we associate each rank-one key–value pair with a scalar weight $w_i \in \mathbb{R}$. For an input token $\mathbf{x} \in \mathbb{R}^{d_{\text{in}}}$ from task $t$, with $r_t$ total ranks, the update becomes

$$\Delta\mathbf{W}^t = \sum\nolimits_{i=1}^{r_t} w_i\mathbf{B}_{:,i}\mathbf{A}_{i,:}, \tag{7}$$

where $w_i$ modulates whether and how strongly each rank $i$ contributes to the updated weight matrix.

### 3.4 Self-Activated Adaptive Sparse Mixture of Ranks

A key challenge for MoE-style CL is routing ambiguity leading to catastrophic forgetting. Conventional routers assign examples to coarse-grained experts, but adding new experts often makes routing suboptimal, causing over-activation or underuse, *i.e.*, the three challenges discussed in Sec. 1. To avoid these pitfalls, we treat each rank-one LoRA (or rank) as an expert based on the rank-level mixture model (Eq. (7)), enabling sparse activation of fine-grained experts for specific tasks. Thus, CL with incrementally added rank-one experts becomes a process of accumulating atomic key–value

knowledge that can be reused in future tasks. Unlike conventional MoE-LoRA (Yang et al., 2024a; Wang et al., 2024; Yu et al., 2024), we use the rank-one $\mathbf{A}_i$'s (keys) as routers to their corresponding $\mathbf{B}_i$'s (values) through self-activation, with each $\mathbf{A}_i$–$\mathbf{B}_i$ pair functioning as a router-expert pair.

**Mixture of self-activated rank-one experts.** Rather than relying on an external router, our framework derives mixture weights for each rank directly from that rank's activation. Recall from Eq. (6) that each rank-one update is parameterized by a key–value memory pair $(\mathbf{A}_{i,:}, \mathbf{B}_{:,i})_{i=1}^{r_t}$; the key $\mathbf{A}_{i,:}$ therefore provides a natural, input-dependent relevance signal. Specifically, given an input token $x \in \mathbb{R}^{d_{\text{in}}}$ from task $t$, and treating $\mathbf{A}$ as the set of keys, we compute a raw score for all $r_t$ ranks accumulated up to task $t$:

$$\mathbf{s}_i = \frac{\mathbf{A}_i x}{\sqrt{\sum_{j=1}^{r_t} \|\mathbf{A}_j x\|_2^2}} \quad \in \mathbb{R}, \tag{8}$$

where the $\ell_2$-normalization across all $r_t$ ranks ensures numerical stability and compensates for activation-scale variations between layers and pre-trained weights. Using raw scores alone matches or surpasses a separately learned router (with 1.5 times parameters) without self-activation (Table 4), showing that each rank can infer its own relevance from activation.

**Self-activated rank selection.** To ensure a stable and gated, sparse activation pattern, we impose a fixed sparse rank-activation budget via `top-k` masking on the $\ell_2$–normalized scores $\mathbf{s}_i := \text{TopK}(\mathbf{s}, k)$, where

$$[\text{TopK}(\mathbf{s}, k)]_i = \begin{cases} \mathbf{s}_i, & \text{if } \mathbf{s}_i \text{ is among the top } k \text{ entries of s,} \\ -\infty, & \text{otherwise,} \end{cases} \tag{9}$$

ensuring that at most $k$ out of the total $r_t$ ranks are eligible to be activated for task $t$. This encourages rank specialization: only a small set of the most relevant ranks are emphasized and trained to capture each kind of input-specific dynamics, and it prevents tiny, noisy activations from affecting gradients and learning dynamics.

**Sharpness control on rank activations.** To further encourage rank specialization and concentrate the update on the most relevant ranks, we temperature-scale and normalize the scores:

$$\mathbf{w}_i = \text{softmax}\left(\frac{s}{\tau_{\text{MoRA}}}\right)_i, \tag{10}$$

where the softmax is taken over the vector $\mathbf{s} = [\mathbf{s}_1, \dots, \mathbf{s}_{r_t}]$ and $\tau_{\text{MoRA}}$ controls the distribution's sharpness. A lower $\tau_{\text{MoRA}}$ concentrates mass on a few ranks (specialists), sharpens the distribution and routes larger learning signals to top ranks, which accelerates specialization and reduces noisy updates to low-relevance ranks, while a higher $\tau_{\text{MoRA}}$ spreads mass (shared representations).

**Threshold-based test-time rank selection.** To further suppress low-signal ranks at inference, we optionally apply a threshold $\delta$ to the $\ell_2$–normalized raw scores and zero out gates below it. We do not hard-threshold during training such that gradients can still flow to near-threshold ranks and allow exploration and adaptation; thresholding only at test time reduces noise and runtime cost:

$$\mathbf{w}_i := \mathbb{1}\{\mathbf{s}_i \geq \delta\} \odot \mathbf{w}_i. \tag{11}$$

This yields a sparse, input-dependent mixture of only the most significant rank-one experts.

Our fine-grained mixture framework offers two key advantages: (1) It reduces *redundancy* and enables *shared-knowledge reuse*: ranks that capture common, cross-task features (such as generic semantics) can be reactivated across multiple tasks, preventing those patterns from being relearned redundantly or interfered; (2) It minimizes *routing ambiguity* and *prevents forgetting*: each rank can specialize on more nuanced patterns, yielding an optimized, rank-wise mixture that integrates new knowledge without disrupting existing representations or hindering future updates.

**Incremental mixture of ranks.** Unlike methods that add a fixed number of LoRA experts per task (Yu et al., 2024), MoRA incrementally adds $r$ new ranks for each task, freezes all prior ranks, and adaptively selects a sparse subset of the most relevant ranks during both training and inference. This design seamlessly incorporates new task-specific knowledge into freshly initialized ranks while preserving earlier ranks unchanged to prevent forgetting. Moreover, it facilitates common knowledge reuse by allowing ranks that encode common patterns to be activated in later tasks.

**Training Objectives.** In our experiments, we only use the model's standard training objective, without any extra regularization or load-balancing constraints. In practice, we empirically observe that the

Table 1: Comparisons on X-TAIL for each domain in terms of "Transfer", "Average", and "Last" scores (%). The **best** and the **second best** results are highlighted in **red** and **blue**, respectively.

| Method | Aircraft | Caltech | DTD | EuroSAT | Flowers | Food | MNIST | OxPet | Cars | SUN397 | Average |
|---|---|---|---|---|---|---|---|---|---|---|---|
| *CLIP* | | | | | | | | | | | |
| Zero-shot | 23.5 | 76.8 | 37.3 | 36.7 | 63.6 | 84.0 | 46.7 | 86.7 | 66.1 | 63.7 | 58.5 |
| Fine-tune | 39.6 | 84.7 | 70.0 | 94.7 | 97.0 | 85.8 | 97.6 | 93.4 | 81.0 | 74.7 | 81.9 |
| *Transfer* | | | | | | | | | | | |
| Zero-shot (Radford et al., 2021) | – | 76.8 | 37.3 | 36.7 | 63.6 | 84.0 | 46.7 | 86.7 | 66.1 | 63.7 | 62.4 |
| LwF (Li & Hoiem, 2017) | – | 66.6 | 26.9 | 19.5 | 51.0 | 78.4 | 26.6 | 68.9 | 35.5 | 56.1 | 47.7 |
| WiSE-FT (Wortsman et al., 2022) | – | 70.1 | 31.9 | 25.3 | 56.3 | 79.8 | 29.9 | 74.9 | 45.6 | 56.8 | 52.3 |
| iCaRL (Rebuffi et al., 2017) | – | 71.7 | 35.0 | 43.0 | 63.4 | 86.9 | 43.9 | 87.8 | 63.7 | 60.0 | 61.7 |
| ZSCL (Zheng et al., 2023) | – | 73.3 | 32.6 | 36.8 | 62.1 | 83.8 | 42.1 | 83.6 | 56.5 | 60.2 | 59.0 |
| MoE-Adapter (Yu et al., 2024) | – | 71.0 | 34.9 | 19.2 | 63.0 | 86.6 | 20.0 | 87.2 | 63.7 | 58.6 | 56.0 |
| RAIL-Primal (Xu et al., 2024) | – | 76.8 | 37.3 | 36.7 | 63.6 | 84.0 | 46.7 | 86.7 | 66.1 | 63.7 | 62.4 |
| CoDyRA (Lu et al., 2024) | – | 74.3 | 36.8 | 44.2 | 69.9 | 83.5 | 42.8 | 88.9 | 64.6 | 63.4 | 63.2 |
| MoRA | – | 74.5 | 38.1 | 46.9 | 65.3 | 82.9 | 45.8 | 88.2 | 65.1 | 62.9 | 63.3 |
| *Average* | | | | | | | | | | | |
| LwF (Li & Hoiem, 2017) | 24.7 | 79.7 | 38.3 | 36.9 | 63.9 | 81.0 | 36.5 | 71.9 | 42.7 | 56.7 | 53.2 |
| WiSE-FT (Wortsman et al., 2022) | 27.1 | 76.5 | 40.9 | 31.3 | 68.7 | 81.6 | 31.4 | 74.7 | 51.7 | 58.4 | 54.2 |
| iCaRL (Rebuffi et al., 2017) | 25.4 | 72.1 | 37.5 | 51.6 | 65.1 | 87.1 | 59.1 | 88.0 | 63.7 | 60.1 | 61.0 |
| ZSCL (Zheng et al., 2023) | 36.0 | 75.0 | 40.7 | 40.5 | 71.0 | 85.3 | 46.3 | 83.3 | 60.7 | 61.5 | 60.0 |
| MoE-Adapter (Yu et al., 2024) | 43.6 | 77.9 | 52.1 | 34.7 | 75.9 | 86.3 | 45.2 | 87.4 | 66.6 | 60.2 | 63.0 |
| RAIL-Primal (Xu et al., 2024) | 42.4 | 89.8 | 55.7 | 68.5 | 84.0 | 83.3 | 65.3 | 85.8 | 67.9 | 64.5 | 70.7 |
| CoDyRA (Lu et al., 2024) | 41.4 | 81.0 | 58.7 | 77.8 | 83.4 | 84.6 | 64.5 | 90.4 | 67.2 | 64.4 | 71.3 |
| MoRA | 44.1 | 81.6 | 64.6 | 79.6 | 83.9 | 84.4 | 66.5 | 89.7 | 68.4 | 64.1 | 72.7 |
| *Last* | | | | | | | | | | | |
| LwF (Li & Hoiem, 2017) | 25.5 | 72.1 | 38.9 | 55.4 | 65.5 | 87.3 | 81.9 | 88.6 | 63.6 | 61.5 | 64.0 |
| WiSE-FT (Wortsman et al., 2022) | 21.8 | 76.8 | 42.9 | 20.8 | 77.5 | 84.9 | 30.7 | 76.6 | 75.8 | 72.5 | 58.0 |
| iCaRL (Rebuffi et al., 2017) | 25.5 | 72.1 | 38.9 | 55.4 | 65.5 | 87.3 | 81.9 | 88.6 | 63.6 | 61.5 | 64.0 |
| ZSCL (Zheng et al., 2023) | 33.1 | 75.3 | 43.5 | 35.2 | 74.6 | 87.4 | 50.4 | 84.2 | 77.3 | 73.4 | 63.4 |
| MoE-Adapter (Yu et al., 2024) | 43.2 | 78.7 | 57.6 | 32.8 | 79.4 | 86.0 | 86.7 | 87.8 | 78.2 | 74.2 | 70.5 |
| RAIL-Primal (Xu et al., 2024) | 41.7 | 94.0 | 66.0 | 86.4 | 97.2 | 82.4 | 93.1 | 83.6 | 75.0 | 71.3 | 79.1 |
| CoDyRA (Lu et al., 2024) | 37.7 | 81.5 | 65.1 | 89.9 | 91.4 | 85.5 | 96.8 | 93.3 | 77.3 | 73.5 | 79.2 |
| MoRA | 37.7 | 81.5 | 70.7 | 92.4 | 95.0 | 86.0 | 97.6 | 92.6 | 81.0 | 74.7 | 80.9 |

conventional load-balancing loss widely employed in the Mixture-of-Experts literature (Shazeer et al., 2017; Lepikhin et al., 2020; Fedus et al., 2022; Dai et al., 2024) is not necessary, thanks to our fine-grained self-activated gating design. Imposing a strict load-balancing constraint might hinder the learning of the key vectors (*i.e.*, matrix $\mathbf{A}$), since each rank should remain free to capture its own characteristic input distribution. Fig. 3 illustrates how each rank learns distinct characteristics.

## 4 EXPERIMENTS

In this paper, we conduct experiments across a diverse set of tasks, including continual learning for both vision-language CLIP models and LMs, and analyze catastrophic forgetting during fine-tuning. Detailed implementation settings and more experiment results are provided in Appendix A.1.

**CL of CLIP.** We evaluate on two benchmarks using CLIP-ViT/B-16 (Radford et al., 2021; Ilharco et al., 2021) on MTIL (Zheng et al., 2023; Yu et al., 2024) and X-TAIL (Xu et al., 2024; Lu et al., 2024), with a range of image classification datasets. Following prior work, we report Transfer (zero-shot on unseen tasks), Last (retention on earlier tasks), and Average (mean accuracy across all datasets of all learning tasks) to assess continual learning performance.

**CL of LM.** We follow prior works (Qin & Joty, 2021; Razdaibiedina et al., 2023; Wang et al., 2023a; Qiao & Mahdavi, 2024) to continually fine-tune T5-large (Raffel et al., 2020) and LLaMA2-7B (Touvron et al., 2023) on five text-classification benchmarks, under three different task orderings. We report the average final accuracy across all tasks after completing the last task.

**Generalization and forgetting on unseen tasks.** To assess effects on pre-trained general knowledge, we fine-tune Llama3.1-8B (Grattafiori et al., 2024) on the CodeAlpaca code-generation dataset (Chaudhary, 2023) and evaluate zero-shot in-domain performance on HumanEval (Chen et al., 2021), as well as out-of-domain accuracy on a broad selection of MMLU (Hendrycks et al., 2021) subjects.

### 4.1 CONTINUAL LEARNING OF CLIP

Table 1 reports X-TAIL results (MTIL results in Appendix Table 11). Prior methods (Yu et al., 2024; Xu et al., 2024) preserve pre-trained capabilities via zero-shot predictions and are therefore capped by the base model's zero-shot performance. Following recent practice (Zheng et al., 2023;

Table 3: Zero-shot in-domain performance on HumanEval (Pass@1) for the code generation domain and out-of-domain accuracy on selected MMLU subjects (formal logic, philosophy, world religions, economics, public relations, STEM, physics, machine learning) after fine-tuning Llama-3.1-8B on CodeAlpaca. The last two columns report trainable parameters (for MoRA: added / activated).

| Method | HumanEval (Pass@1) | Out-of-Domain (Acc.) | | | | | | | | | Params (M) | %Params |
|---|---|---|---|---|---|---|---|---|---|---|---|---|
| | | Logic | Phil. | Reli. | Econ. | Pub. Rel. | STEM | Phys. | ML | MMLU | | |
| Llama-3.1-8B | 38.40 | 42.06 | **71.06** | **83.63** | 70.17 | **68.18** | 54.84 | 39.22 | **40.18** | 63.45 | — | — |
| LoRA ($r = 4$) | 41.46 | 39.68 | 70.09 | 81.87 | 71.43 | 65.45 | 54.77 | **45.10** | 40.17 | 63.28 | 10.5 | 0.13% |
| LoRA ($r = 8$) | 44.51 | 39.68 | **70.74** | 81.87 | 71.85 | 64.54 | 54.17 | 42.16 | 39.29 | 63.03 | 21.0 | 0.26% |
| LoRA ($r = 16$) | **45.73** | 41.27 | 68.49 | 80.70 | 72.69 | **66.36** | 54.96 | 44.11 | 38.39 | 63.35 | 41.9 | 0.52% |
| LoRA ($r = 32$) | **47.56** | 42.85 | 69.45 | 81.87 | 72.27 | **66.36** | **55.44** | 45.10 | 39.29 | **63.59** | 83.9 | 1.03% |
| MoRA | **47.56** | 48.41 | 70.09 | 82.46 | 73.53 | 68.18 | 55.53 | 46.08 | 41.96 | 63.70 | 41.9/26.2 | 0.52%/**0.32%** |

Lu et al., 2024), MoRA continuously train the pre-trained model so it can surpass that zero-shot ceiling and better generalize to unseen domains, while still preserving existing capabilities. Across both Last and Average metrics, MoRA substantially outperforms prior works. Unlike methods that rely on domain-ID prediction (Yu et al., 2024), high-dimensional projection layers or feature-banks (Xu et al., 2024), MoRA improves the pre-trained representation without external mechanisms; compared with (Lu et al., 2024), which learns fixed weights per rank-$r$ update, MoRA uses input-dependent gates on individual rank-one updates for finer expert specialization and dynamic utilization.

## 4.2 CONTINUAL LEARNING OF LM

Table 2 reports results across three task orderings: MoRA consistently outperforms prior methods and closely approaches the multi-task learning (MTL) upper bound. Unlike O-LoRA (Wang et al., 2023a) and LB-CL (Qiao & Mahdavi, 2024), which rely on orthogonality constraints or gradient projections between per-task LoRA adapters (potentially limiting adapter capacity), MoRA needs no extra regularization. By decomposing each rank-r update into rank-one components and applying self-activated, sparse gating, MoRA lets each component specialize on its own input distribution, reducing interference and more effectively capturing diverse patterns. We further evaluate MoRA on LLaMA2-7B (Touvron et al., 2023) under the same continual-learning setup (Table 9 in the Appendix), MoRA also outperforms O-LoRA by 2.3% averaged over 3 task orders.

Table 2: Summary of results on standard CL benchmarks with T5-large. We report averaged accuracy after training on the last task across three task orderings.

| Method | Standard CL Benchmark | | | |
|---|---|---|---|---|
| | Order-1 | Order-2 | Order-3 | *Avg.* |
| MTL | 80.0 | | | |
| SeqFT | 18.9 | 24.9 | 41.7 | 28.5 |
| SeqLoRA | 44.6 | 32.7 | 53.7 | 43.7 |
| IncLoRA | 66.0 | 64.9 | 68.3 | 66.4 |
| Replay | 55.2 | 56.9 | 61.3 | 57.8 |
| EWC | 48.7 | 47.7 | 54.5 | 50.3 |
| LwF | 54.4 | 53.1 | 49.6 | 52.3 |
| L2P | 60.3 | 61.7 | 61.1 | 60.7 |
| LFPT5 | 67.6 | 72.6 | 77.9 | 72.7 |
| InfLoRA | 75.2 | 75.4 | 75.8 | 75.5 |
| O-LoRA | 75.4 | 75.7 | 76.3 | 75.8 |
| LB-CL | 76.9 | 76.5 | 76.8 | 76.7 |
| MoRA | **77.4** | **77.5** | **77.9** | **77.6** |

## 4.3 FORGETTING AND GENERALIZATION DURING FINE-TUNING

Table 3 examines how standard fine-tuning affects both in-domain performance and out-of-domain generalization. We observe two key phenomena. *(1) Forgetting of unrelated knowledge:* conventional LoRA fine-tuning causes drops on topics that are unrelated to the fine-tuning data (e.g., Philosophy, World Religions, Public Relations), indicating that pre-trained general knowledge is overwritten during adaptation. *(2) Improved generalization*: fine-tuning on code data increases accuracy on logically and quantitatively oriented tasks (Formal Logic, Economics, STEM, Physics, Machine Learning), showing that representations learned from code transfer to related symbolic domains.

In contrast, MoRA both preserves unrelated pre-trained knowledge and improves in- and out-of-domain generalization. Decomposing each low-rank update into specialized rank-one components and activating only the most relevant subset prevents catastrophic overwriting of prior knowledge while allowing targeted adaptation. At the same time, MoRA achieves strong code-generation performance using roughly one-third of the activated parameters required by a rank-32 LoRA and substantially improves out-of-domain accuracy.

## 4.4 VISUALIZATIONS OF FINE-GRAINED RANK ACTIVATIONS

Figure 3 shows rank activations recorded during continual learning. These visualizations illustrate two properties of MoRA: (1) individual ranks specialize on distinct input patterns, and (2) the model

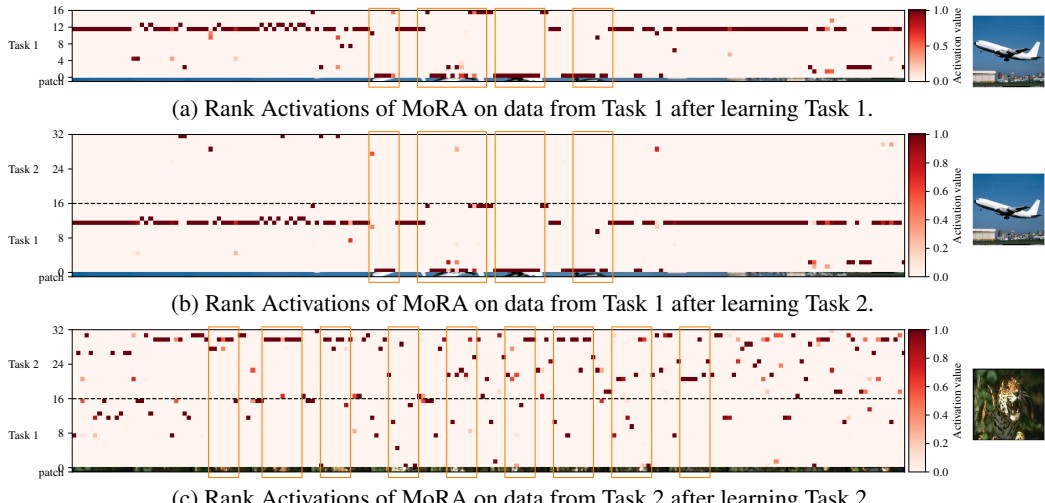

(a) Rank Activations of MoRA on data from Task 1 after learning Task 1.

(b) Rank Activations of MoRA on data from Task 1 after learning Task 2.

(c) Rank Activations of MoRA on data from Task 2 after learning Task 2.

Figure 3: Visualization of MoRA rank activations during Task 1 and Task 2 training. Activations are extracted from the K projection in the attention module (layer 8) of the image encoder. Corresponding image patches are shown below each activation map, with regions relevant to each class marked by orange bounding boxes. Zoom in for details. More visualizations are in Figs. 6 and 7 of the Appendix, demonstrating forgetting mitigation and knowledge reuse.

substantially reduces cross-task interference, thereby mitigating forgetting. Extended visualizations across more tasks and scenarios appear in Fig. 6 and Fig. 7 in the Appendix.

**Each rank specializes in distinct input patterns.** In Fig. 3a, airplane patches strongly activate Rank 0, while blue-sky backgrounds predominantly activate Rank 11. In Fig. 3c, Ranks 19, 20, and 29 (orange boxes) respond to jaguar patches. The more complex backgrounds in Fig. 3c (*e.g.*, leaves, shadows) yield a richer, more distributed pattern than the simple blue sky in Fig. 3a, highlighting MoRA's capacity to model contextual complexity. Some ranks learned earlier are also reused on later tasks (Fig. 7, Appendix), indicating transfer of shared semantics.

**Reduced cross-task interference and mitigated forgetting.** Comparing the same input after Task 1 (Fig. 3a) and after Task 2 (Fig. 3b) shows almost identical activations (more in Fig. 6, Appendix): Rank 0 still responds to airplane semantics and Rank 11 to blue-sky patches. This stability indicates that our self-activated, sparse mixture-of-ranks prevents later updates from overwriting earlier task representations, reducing inter-task interference and mitigating catastrophic forgetting.

Table 4: Routing/activation strategy

| Mixture Calculation | Transfer | Average | Last |
|---|---|---|---|
| MoE-LoRA | 62.56 | 69.45 | 74.53 |
| Non-self-activation | 60.09 | 65.97 | 69.76 |
| Self-activation | 60.26 | 65.94 | 69.85 |
| *w/* top-k rank selection | 60.69 | 66.52 | 70.62 |
| *w/* sharpness control with $\tau_{\text{MoRA}}$ | 62.07 | 71.15 | 79.62 |
| *w/* test-time rank thresholding | 60.78 | 66.83 | 71.08 |
| **MoRA** | **63.30** | **72.70** | **80.90** |

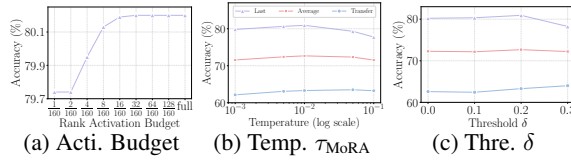

(a) Acti. Budget   (b) Temp. $\tau_{\text{MoRA}}$   (c) Thre. $\delta$

Figure 4: Ablation on (a) rank activation budget, (b) temperature $\tau_{\text{MoRA}}$, and (c) threshold $\delta$.

### 4.5 ABLATION STUDIES

**Routing/activation strategy.** Table 4 compares different strategies for computing mixture weights, under a controlled setup with one LoRA per task, standard training loss, and no additional regularization: (1) *MoE-LoRA* trains a router to assign weights to each LoRA adapter; its coarse-grained design and lack of mechanisms that encourage expert specialization lead to greater interference and increased forgetting compared to MoRA. (2) *Non-self-activation* trains a finer-grained router for mixing each rank-one component. However, with a much larger number of experts (ranks), the router becomes increasingly prone to confusion, resulting in higher training complexity and degraded performance compared with MoE-LoRA. (3) *Self-activation* infers weights directly from softmax-normalized activations (Eq. 8), matching or exceeding Non-self-activation and demonstrating that each rank can infer its own relevance without extra trainable router modules. (4) *Top-k rank selection* restricts updates to the k most relevant ranks, reducing interference and improving accuracy. (5) *Sharpness control*

concentrates activation mass and substantially boosts specialization. (6) *Test-time rank thresholding* removes low-importance ranks at inference, yielding a slight gain by eliminating noisy contributions.

**Rank activation budget.** We probe how many ranks are necessary by varying the activation budget (Fig. 4a). To isolate this effect, we disable other sparsity mechanisms (*e.g.*, thresholding). After training on 10 tasks (160 ranks total), increasing the budget from 2 (1.25%) to 16 active ranks (10%) yields a large improvement in final accuracy and then gradually plateaus. This suggests that each rank specializes in distinct input patterns and that a small, sparse subset of ranks is sufficient to cover diverse inputs. Even with a high budget, the self-activating gating remains robust: it still privileges the most relevant ranks per token while suppressing lower-relevance contributions.

**Sharpness control of $\tau_{\mathbf{MoRA}}$.** We use temperature scaling (Eq. (10)) to control mixture sharpness: lower $\tau_{MoRA}$ yields sharper activations, while higher $\tau_{MoRA}$ flattens the distribution and engages more ranks. Fig. 4b shows that increasing $\tau_{MoRA}$ improves transfer to unseen tasks, even surpassing the reported state-of-the-art Transfer in Table 1 when $\tau_{MoRA}$ is high. A moderate $\tau_{MoRA}$ around 0.01 balances specificity and generalization, delivering strong Last and Average scores.

**Threshold-based rank selection.** Figure 4c shows the effect of the test-time rank selection threshold $\delta$. Applying a modest threshold removes low-activation ranks and reduces noisy contributions, which improves both downstream adaptation and out-of-domain generalization.

## 5 CONCLUSION

We present MoRA, a self-activated sparse mixture-of-ranks framework that decomposes each LoRA update into rank-1 experts and self-infer their sparse mixture weights. By reducing knowledge redundancy and resolving router–expert mismatch during CL, MoRA delivers superior continual-learning performance, significantly reduces forgetting, and enhances generalization.

**Future works.** MoRA controls mixture sharpness and sparsity via top-k selection, temperature scaling, and thresholding. While effective, the sparse activation mechanism can be further improved by incorporating awareness of the input data distribution. Learning this temperature or adapting the mixture's sharpness based on data offers a promising avenue for future research. We also aim to extend the method's applicability to a broader range of PTMs and applications.

REPRODUCIBILITY STATEMENT

Further implementation details and dataset descriptions are provided in Sec. 4 and Appendix A.1. Our source code will be publicly released upon paper acceptance.

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

THE USE OF LARGE LANGUAGE MODELS (LLMS)

Large language models (LLMs) were employed in preparing this manuscript, specifically to assist with grammar checking and polishing. All technical content, experimental design, and analysis were conceived and carried out by the authors.

# A    ADDITIONAL DETAILS AND RESULTS OF EXPERIMENTS

Table 5: Details of the 15 datasets used in our continual-learning experiments using LMs. NLI denotes natural language inference, and QA denotes question-answering tasks. The first five tasks comprise the standard CL benchmark; the remaining ten tasks are used for the extended long-sequence evaluations.

| Dataset name | Category | Task | Domain | Metric |
|---|---|---|---|---|
| 1. Yelp | CL Benchmark | sentiment analysis | Yelp reviews | accuracy |
| 2. Amazon | CL Benchmark | sentiment analysis | Amazon reviews | accuracy |
| 3. DBpedia | CL Benchmark | topic classification | Wikipedia | accuracy |
| 4. Yahoo | CL Benchmark | topic classification | Yahoo Q&A | accuracy |
| 5. AG News | CL Benchmark | topic classification | news | accuracy |
| 6. MNLI | GLUE | NLI | various | accuracy |
| 7. QQP | GLUE | paragraph detection | Quora | accuracy |
| 8. RTE | GLUE | NLI | news, Wikipedia | accuracy |
| 9. SST-2 | GLUE | sentiment analysis | movie reviews | accuracy |
| 10. WiC | SuperGLUE | word sense disambiguation | lexical databases | accuracy |
| 11. CB | SuperGLUE | NLI | various | accuracy |
| 12. COPA | SuperGLUE | QA | blogs, encyclopedia | accuracy |
| 13. BoolQA | SuperGLUE | boolean QA | Wikipedia | accuracy |
| 14. MultiRC | SuperGLUE | QA | various | accuracy |
| 15. IMDB | SuperGLUE | sentiment analysis | movie reviews | accuracy |

Table 6: Instructions for different tasks.

| Task | Prompts |
|---|---|
| NLI | What is the logical relationship between the "sentence 1" and the "sentence 2"? Choose one from the option. |
| QQP | Whether the "first sentence" and the "second sentence" have the same meaning? Choose one from the option. |
| SC | What is the sentiment of the following paragraph? Choose one from the option. |
| TC | What is the topic of the following paragraph? Choose one from the option. |
| BoolQA | According to the following passage, is the question true or false? Choose one from the option. |
| MultiRC | According to the following passage and question, is the candidate answer true or false? Choose one from the option. |
| WiC | Given a word and two sentences, whether the word is used with the same sense in both sentence? Choose one from the option. |

## A.1    DETAILED EXPERIMENT SETTINGS

**CL on CLIP.** We follow the experimental setups in (Zheng et al., 2023; Yu et al., 2024; Xu et al., 2024) and use the CLIP model with a ViT-B/16 backbone (Radford et al., 2021) for all experiments. By default, MoRA is applied to every pre-trained weight matrix in both the vision and text encoders, with an initial rank of 16 per update. Each task is trained for 500 iterations using AdamW (Loshchilov & Hutter, 2017) with a learning rate of $5e-4$. During continual learning, we freeze the ranks learned from previous tasks and initialize new $r = 16$ ranks for each incoming task. The rank activation budget is set to 16 throughout all tasks. We set the temperature $\tau_{\text{MoRA}} = 0.1$ and the threshold $\delta = 0.2$.

Table 7: The six task-sequence orders used in our continual learning experiments. Sequences 1–3 follow the standard CL benchmarks employed in prior work. Sequences 4–6 extend to longer 15-task streams, as introduced in (Razdaibiedina et al., 2023).

| Order | Task Sequence |
|---|---|
| 1 | dbpedia → amazon → yahoo → ag |
| 2 | dbpedia → amazon → ag → yahoo |
| 3 | yahoo → amazon → ag → dbpedia |
| 4 | mnli → cb → wic → copa → qqp → boolqa → rte → imdb → yelp → amazon → sst-2 → dbpedia → ag → multirc → yahoo |
| 5 | multirc → boolqa → wic → mnli → cb → copa → qqp → rte → imdb → sst-2 → dbpedia → ag → yelp → amazon → yahoo |
| 6 | yelp → amazon → mnli → cb → copa → qqp → rte → imdb → sst-2 → dbpedia → ag → yahoo → multirc → boolqa → wic |

**CL on LMs.** We follow the protocol of previous work in continually fine-tuning the T5-large (Raffel et al., 2020) and LLaMA2-7B (Touvron et al., 2023) on a suite of text-classification tasks. We train on five standard benchmarks—AG News, Amazon Reviews, Yelp Reviews, DBpedia, and Yahoo Answers—using three distinct task orderings drawn from (Qin & Joty, 2021; Razdaibiedina et al., 2023; Wang et al., 2023a; Qiao & Mahdavi, 2024). To probe longer sequences, we extend this to a 15-dataset stream (Table 5), incorporating tasks from the original CL benchmark (Zhang et al., 2015), GLUE (Wang et al., 2018), SuperGLUE (Wang et al., 2019), and the IMDB movie reviews corpus. Natural language prompts for each task are presented in Table 6, with NLI tasks (MNLI, RTE, CB), sentiment classification (Amazon, Yelp, SST-2, IMDB), and topic classification (AG News, DBpedia, Yahoo).

We evaluate three distinct task sequences for both the standard CL and 15-task benchmarks (Table 7). After completing the final task in each stream, we report the average accuracy across all tasks. All experiments use one epoch per task with DeepSpeed, a fixed learning rate of $1e-3$, batch size 64, and dropout of 0.1. MoRA is applied to both the query and key projection matrices within attention layers, initializing $r = 8$ new ranks for each incoming task similarly as in (Wang et al., 2023a; Qiao & Mahdavi, 2024). We maintain a constant activation budget of 4 ranks throughout continual learning, set the temperature $\tau_{\text{MoRA}} = 0.1$, and the threshold $\delta = 0.2$.

**Generalization and forgetting on unseen tasks.** To assess effects on pre-trained general knowledge, we fine-tune Llama3.1-8B (Grattafiori et al., 2024) on the CodeAlpaca code-generation dataset (Chaudhary, 2023) using llama-Factory (Zheng et al., 2024) and evaluate using lm-eval-harness (Gao et al., 2024) on zero-shot in-domain performance on HumanEval (Chen et al., 2021), as well as out-of-domain accuracy on a broad selection of MMLU (Hendrycks et al., 2021) subjects—Formal Logic, Philosophy, World Religions, Economics, Public Relations, STEM, Physics, and Machine Learning.

In this experiment, MoRA is applied to all linear weight matrices of the pre-trained model. We fine-tune on CodeAlpaca with a batch size of 32 over 3 epochs and a cosine learning-rate schedule starting at $5e-4$. We train with $r = 16$ ranks and enforce a constant activation budget of 4 ranks. The self-routed gating uses a temperature $\tau_{\text{MoRA}} = 0.5$ and a threshold $\delta = 0.2$. We observe that, due to variations in hidden representations across architectures, the optimal temperature setting can differ across each pre-trained models.

**Further details of the used datasets in MTIL and X-TAIL.** The MTIL setting consists of 1,201 classes drawn from 11 diverse datasets: Aircraft (Maji et al., 2013), Caltech101 (Fei-Fei et al., 2004), CIFAR100 (Krizhevsky et al., 2009), DTD (Cimpoi et al., 2014), EuroSAT (Helber et al., 2019), Flowers (Nilsback & Zisserman, 2008), Food (Bossard et al., 2014), MNIST (Deng, 2012), OxfordPet (Parkhi et al., 2012), Cars (Krause et al., 2013), and SUN397 (Xiao et al., 2010). In the X-TAIL setting, a total of 10 datasets are used, with CIFAR100 (Fei-Fei et al., 2004) excluded to prevent domain overlap, following the protocol in (Xu et al., 2024). In line with (Xu et al., 2024), we use a 5-shot split for MTIL and a 16-shot split for X-TAIL.

Table 8: Average accuracy on T5-large continual-learning benchmarks after the final task, evaluated over extended 15-task sequences. Results for prior methods are taken from (Qiao & Mahdavi, 2024).

| Method | Large Number of Tasks | | | |
| --- | --- | --- | --- | --- |
| | Order-4 | Order-5 | Order-6 | *Avg.* |
| MTL | 76.5 | | | |
| SeqFT | 7.4 | 7.4 | 7.5 | 7.4 |
| SeqLoRA | 2.3 | 0.6 | 1.9 | 1.6 |
| IncLoRA | 63.3 | 58.5 | 61.7 | 61.2 |
| Replay | 55 | 54.6 | 53.1 | 54.2 |
| EWC | 45.3 | 44.5 | 45.6 | 45.1 |
| LwF | 50.1 | 43.1 | 47.4 | 46.9 |
| L2P | 57.5 | 53.8 | 56.9 | 56.1 |
| LFPT5 | 69.8 | 67.2 | 69.2 | 68.7 |
| O-LoRA | **70.5** | 65.5 | 70.5 | 68.8 |
| LB-CL | 68.4 | 67.3 | 71.8 | 69.2 |
| MoRA | 68.91 | **68.32** | **71.95** | **69.72** |

Table 9: Continual learning results on standard CL benchmarks with the LLaMA2-7B model.

| Method | Order-1 | Order-2 | Order-3 | *Avg.* |
| --- | --- | --- | --- | --- |
| O-LoRA | 76.8 | 75.7 | 75.7 | 76.1 |
| MoRA | **77.8** | **78.0** | **79.3** | **78.4** |

## A.2 CONTINUAL LEARNING OF LMS ON LONG TASK SEQUENCES

In Sec. 4.2 Table 2, we evaluate MoRA on standard CL benchmarks (Razdaibiedina et al., 2023). In Table 8, we extend the evaluation to challenged long task sequences using 15 datasets with 3 different orderings as in (Wang et al., 2023a; Qiao & Mahdavi, 2024). Consistent with the findings in Table 2, MoRA outperforms previous methods in terms of averaged performance across three task orders, and largely close the gap to multi-task learning. Two key design choices drive this robust performance in the long-sequence regime. First, by decomposing each LoRA update into fine-grained rank-1 components and enforcing a small, fixed activation budget, MoRA encourages each rank to specialize on a narrow subspace of the data manifold. At inference time, only the most relevant subspaces are activated for a given input, which preserves earlier task representations and prevents catastrophic interference. Second, our self-routed gating mechanism enables each rank to assess its own relevance on a per-token basis, yielding stable mixture patterns as the expert pool grows. Coupling with our proposed rank pruning, these mechanisms ensure that MoRA continually incorporates new knowledge only when needed while robustly maintaining prior capabilities.

Table 10: Comparisons of trainable parameters for each pre-trained weight matrix during continual learning of each task. MoE-LoRA and MoE-Adapter trains additional router module with LoRA experts. In MoRA, $k$ denotes the rank activation budget (with $k \leq r$).

| Method | Trainable parameters per task |
| --- | --- |
| LoRA (Hu et al., 2021) | $r\,(d_{\text{in}} + d_{\text{out}})$ |
| MoE-LoRA (1 expert/task) | $r\,(d_{\text{in}} + d_{\text{out}}) + d_{\text{in}}$ |
| MoE-Adapter (2 experts/task) (Yu et al., 2024) | $2(r\,(d_{\text{in}} + d_{\text{out}}) + d_{\text{in}})$ |
| CoDyRA (Lu et al., 2024) | $r\,(d_{\text{in}} + d_{\text{out}}) + r$ |
| O-LoRA (Wang et al., 2023a) | $r\,(d_{\text{in}} + d_{\text{out}})$ |
| LB-CL (Qiao & Mahdavi, 2024) | $r\,(d_{\text{in}} + d_{\text{out}}) + r$ |
| MoRA | $rd_{\text{in}} + kd_{\text{out}}$ |

## A.3 MULTI-DOMAIN TASK INCREMENTAL LEARNING.

We evaluate MoRA in the few-shot MTIL setting (Table 11) under the same protocols as (Yu et al., 2024; Xu et al., 2024; Lu et al., 2024). Consistent with the results observed in the X-TAIL setting, our method demonstrates clear superiority in this scenario.

In this challenging scenario, the model must learn 11 diverse tasks sequentially, with only five examples per class. These findings validate that our self-activated sparse mixture-of-ranks framework both facilitates continual acquisition of new knowledge and mitigates forgetting from the pre-trained model and earlier tasks.

Table 11: Comparisons on 5-shot MTIL setting. Following the same protocol as in (Yu et al., 2024; Xu et al., 2024; Lu et al., 2024).

| Method | Aircraft | Caltech101 | CIFAR100 | DTD | EuroSAT | Flowers | Food | MNIST | OxfordPet | Cars | SUN397 | Average |
|---|---|---|---|---|---|---|---|---|---|---|---|---|
| *CLIP* | | | | | | | | | | | | |
| Zero-shot (Radford et al., 2021) | 24.3 | 88.4 | 68.2 | 44.6 | 54.9 | 71.0 | 88.5 | 59.4 | 89.0 | 64.7 | 65.2 | 65.3 |
| *Transfer* | | | | | | | | | | | | |
| Zero-shot (Radford et al., 2021) | – | 88.4 | 68.2 | 44.6 | 54.9 | 71.0 | 88.5 | 59.6 | 89.0 | 64.7 | 65.2 | 69.4 |
| LwF (Li & Hoiem, 2017) | – | 72.1 | 49.2 | 35.9 | 44.5 | 41.1 | 66.6 | 50.5 | 69.0 | 19.0 | 51.7 | 50.0 |
| LwF-VR (Ding et al., 2022) | – | 82.2 | 62.5 | 40.1 | 40.1 | 56.3 | 80.0 | 60.9 | 77.6 | 40.5 | 60.8 | 60.1 |
| WiSE-FT (Wortsman et al., 2022) | – | 77.6 | 60.0 | 41.3 | 39.4 | 53.0 | 76.6 | 58.1 | 75.5 | 37.3 | 58.2 | 57.7 |
| ZSCL (Zheng et al., 2023) | – | 84.0 | 68.1 | 44.8 | 46.8 | 63.6 | 84.9 | 61.4 | 81.4 | 55.5 | 62.2 | 65.3 |
| MoE-Adapter (Yu et al., 2024) | – | 87.9 | 68.2 | 44.1 | 48.1 | 64.7 | 88.8 | 69.0 | 89.1 | 64.5 | 65.1 | 68.9 |
| RAIL-Primal (Xu et al., 2024) | – | 88.4 | 68.2 | 44.6 | 54.9 | 71.0 | 88.5 | 59.6 | 89.0 | 64.7 | 65.2 | 69.4 |
| CoDyRA (Lu et al., 2024) | – | 92.4 | 68.4 | 45.8 | 54.5 | 69.6 | 87.4 | 65.2 | 88.5 | 64.2 | 64.5 | 69.9 |
| MoRA | – | 92.0 | 68.8 | 45.6 | 53.1 | 68.6 | 84.4 | 64.3 | 89.8 | 65.4 | 64.8 | 69.7 |
| *Average* | | | | | | | | | | | | |
| LwF (Li & Hoiem, 2017) | 23.5 | 77.4 | 43.5 | 41.7 | 43.5 | 52.2 | 54.6 | 63.4 | 68.0 | 21.3 | 52.6 | 49.2 |
| LwF-VR (Ding et al., 2022) | 24.9 | 89.1 | 64.2 | 53.4 | 54.3 | 70.8 | 79.2 | 66.5 | 79.2 | 44.1 | 61.6 | 62.5 |
| WiSE-FT (Wortsman et al., 2022) | 32.0 | 87.7 | 61.0 | 55.8 | 68.1 | 69.3 | 76.8 | 71.5 | 77.6 | 42.0 | 59.3 | 63.7 |
| ZSCL (Zheng et al., 2023) | 28.2 | 88.6 | 66.5 | 53.5 | 56.3 | 73.4 | 83.1 | 56.4 | 82.4 | 57.5 | 62.9 | 64.4 |
| MoE-Adapter (Yu et al., 2024) | 30.0 | 89.6 | 73.9 | 58.7 | 69.3 | 79.3 | 88.1 | 76.5 | 89.1 | 65.3 | 65.8 | 71.4 |
| RAIL-Primal (Xu et al., 2024) | 32.9 | 94.5 | 69.9 | 58.1 | 71.8 | 84.4 | 88.5 | 70.4 | 89.0 | 66.1 | 65.7 | 71.9 |
| CoDyRA (Lu et al., 2024) | 34.6 | 95.8 | 73.9 | 60.0 | 77.1 | 81.3 | 86.6 | 75.9 | 89.9 | 66.1 | 65.3 | 73.3 |
| MoRA | 36.7 | 95.4 | 74.9 | 61.9 | 77.1 | 82.6 | 85.3 | 76.0 | 90.5 | 67.0 | 65.6 | 73.9 |
| *Last* | | | | | | | | | | | | |
| LwF (Li & Hoiem, 2017) | 22.1 | 58.2 | 17.9 | 32.1 | 28.1 | 66.7 | 46.0 | 84.3 | 64.1 | 31.5 | 60.1 | 46.5 |
| LwF-VR (Ding et al., 2022) | 22.9 | 89.8 | 59.3 | 57.1 | 57.6 | 79.2 | 78.3 | 77.7 | 83.6 | 60.1 | 69.8 | 66.9 |
| WiSE-FT (Wortsman et al., 2022) | 30.8 | 88.9 | 59.6 | 60.3 | 80.9 | 81.7 | 77.1 | 94.9 | 83.2 | 62.8 | 70.0 | 71.9 |
| ZSCL (Zheng et al., 2023) | 26.8 | 88.5 | 63.7 | 55.7 | 60.2 | 82.1 | 82.6 | 58.6 | 85.9 | 66.7 | 70.4 | 67.4 |
| MoE-Adapter (Yu et al., 2024) | 30.1 | 89.3 | 74.9 | 64.0 | 82.3 | 89.4 | 87.1 | 89.0 | 89.1 | 69.5 | 72.5 | 76.1 |
| RAIL-Primal (Xu et al., 2024) | 32.9 | 95.1 | 70.3 | 63.2 | 81.5 | 95.6 | 88.5 | 89.7 | 89.0 | 72.5 | 71.0 | 77.2 |
| CoDyRA (Lu et al., 2024) | 31.6 | 95.5 | 72.8 | 63.5 | 85.0 | 89.7 | 85.0 | 94.7 | 93.2 | 73.6 | 73.0 | 78.0 |
| MoRA | 32.5 | 95.3 | 75.3 | 66.6 | 87.8 | 92.6 | 86.3 | 96.3 | 92.6 | 73.5 | 73.8 | 79.3 |

## A.4 MORE COMPARISONS ON CL ON LMS

**Results on TRACE benchmark.** To evaluate our method on Large Language Models (LLMs), we use the TRACE dataset (Wang et al., 2023b), a benchmark specifically designed for continual-learning research in LLMs. On this benchmark, we experiment with LLaMA-3.2-1B-Instruct (Dubey et al., 2024) and report both Overall Performance (OP) and Backward Transfer (BWT). As shown in Table 12, MoRA performs robustly on LLMs (LLaMA-3.2-1B-Instruct) and achieves results competitive with existing methods.

**Results on SuperNI benchmark.** To provide a complete comparison, we additionally evaluate MoRA against SAPT (Zhao et al., 2024) on the SuperNI benchmark (Wang et al., 2022d) using the T5-Large backbone (Table 13). We report results in both replay-based and replay-free settings. Under the replay-based protocol introduced by SAPT, where pseudo-samples are generated using a trained generative model, MoRA attains a stronger stability–plasticity trade-off (AP: 51.79% vs. 51.54%; FT: 0.73% vs. 0.91%). In the replay-free setting, which is the primary use case for MoRA, our method achieves state-of-the-art performance (AP: 39.62%) and substantially outperforms parameter-efficient baselines such as O-LoRA (26.37%) and L2P (15.18%). These results highlight the complementary design choices. SAPT improves replay efficiency through shared attention prompts, while MoRA relies on architectural isolation via sparse rank-1 experts. As a result, MoRA remains effective without a memory buffer, yet can also incorporate replay to further improve performance beyond SAPT.

**Discussions on performance robustness.** To assess order robustness, we incorporate the OPD metric proposed by (Yoon et al., 2020), which measures a model's sensitivity to the sequence of arriving tasks. Following standard practice for evaluating global performance stability, we compute the standard deviation of the final average accuracy over the $K$ task orders considered. In Table 14, using results on the Standard CL Benchmark with three distinct task orders, MoRA demonstrates substantially improved robustness to task ordering. The disparity across orders is 0.26 for MoRA, which is approximately half of that of O-LoRA (0.46). This indicates that the Self-Activated Sparse Mixture mechanism effectively reduces task interference and mitigates the unidirectional knowledge transfer effects identified in prior work.

Table 12: Comparison with a broad range of CL methods on the TRACE benchmark using the LLaMA-3.2-1B-Instruct backbone. We report Overall Performance (OP (%) ↑) and Backward Transfer (BWT (%) ↓). Results are averaged over three runs with standard deviations. The best results are highlighted in bold.

|  | FIX(ICL) | SeqLoRA | OGD | GEM | EWC | L2P | DualPrompt | HiDeLoRA | O-LoRA | TreeLoRA | MoRA |
|---|---|---|---|---|---|---|---|---|---|---|---|
| OP | $31.16_{\pm0.4}$ | $29.73_{\pm1.6}$ | $30.12_{\pm2.0}$ | $32.19_{\pm2.0}$ | $31.96_{\pm1.6}$ | $29.38_{\pm1.2}$ | $30.76_{\pm1.2}$ | $33.73_{\pm1.2}$ | $32.94_{\pm0.8}$ | $36.14_{\pm0.7}$ | $\mathbf{37.77_{\pm0.8}}$ |
| BWT | – | $17.03_{\pm1.2}$ | $15.2_{\pm1.6}$ | $10.74_{\pm1.6}$ | $11.62_{\pm1.2}$ | $13.57_{\pm0.8}$ | $11.34_{\pm0.8}$ | $12.36_{\pm0.8}$ | $12.89_{\pm1.2}$ | $7.36_{\pm0.8}$ | $\mathbf{3.12_{\pm0.8}}$ |

Table 13: Overall results on the SuperNI Benchmark using the T5-Large backbone. We report Average Performance (AP, ↑) and Forgetting (FT, ↓). The best results for the stability-plasticity trade-off are highlighted in bold for methods with and without memory replay, respectively.

| Methods | Replay | SuperNI | |
|---|---|---|---|
|  |  | AP | FT |
| *Replay-Based Methods* | | | |
| Replay | ✓ | 35.37 | 16.92 |
| SAPT | ✓ | 51.54 | 0.91 |
| MoRA | ✓ | **51.79** | **0.73** |
| Replay-Free Methods | | | |
| L2P | ✗ | 15.18 | 3.65 |
| IncLoRA | ✗ | 12.33 | 41.93 |
| C-LoRA | ✗ | 22.69 | 24.25 |
| O-LoRA | ✗ | 26.37 | 19.15 |
| MoRA | ✗ | **39.62** | **5.74** |

Table 14: Order Robustness Analysis on standard CL benchmark with T5-Large. We report the accuracy for each order and the average accuracy ± the standard deviation.

| Method | Order 1 | Order 2 | Order 3 | Avg ± Std |
|---|---|---|---|---|
| SeqFT | 18.9 | 24.9 | 41.7 | 28.5 ± 11.82 |
| L2P | 60.3 | 61.7 | 61.1 | 61.0 ± 0.70 |
| LFPT5 | 67.6 | 72.6 | 77.9 | 72.7 ± 5.15 |
| O-LoRA | 75.4 | 75.7 | 76.3 | 75.8 ± 0.46 |
| MoRA | **77.4** | **77.5** | **77.9** | **77.6 ± 0.26** |

## A.5 MORE COMPARISON ON HUMANEVAL FOR CODE GENERATION

To further evaluate code generation performance, we compare MoRA against LoRI-D and LoRI-S (Zhang et al., 2025) on the HumanEval benchmark (Table 15). MoRA consistently outperforms both LoRI variants across all metrics, achieving notable improvements in Pass@1, Pass@5, and especially Pass@10.

Table 15: Performance comparison on the HumanEval benchmark, reported in terms of Pass@1, Pass@5, and Pass@10.

| HumanEval | Pass@1 | Pass@5 | Pass@10 |
|---|---|---|---|
| LoRI-D | 43.2 | 57.6 | 63.2 |
| LoRI-S | 41.3 | 54.4 | 59.6 |
| MoRA | **47.6** | **60.9** | **70.1** |

## A.6 PERFORMANCE ROBUSTNESS

MoRA employs a sparse mixture of previously learned and newly introduced rank-1 experts to capture both shared and task-specific knowledge, resulting in substantially improved Last performance. To assess statistical significance and robustness, we report mean and standard deviation over three independent runs (Table 16). MoRA consistently outperforms competing methods across all metrics and exhibits lower variance, highlighting its effectiveness and stability in continual-learning scenarios.

Table 16: Comparison to InFLoRA and performance robustness. We report mean and standard deviation across 3 independent runs. Best performances are marked in **bold**.

| Method | Cars | Aircraft | OxfordPet | Food | SUN397 | MNIST | Flowers | DTD | Caltech101 | EuroSAT | Average |
|---|---|---|---|---|---|---|---|---|---|---|---|
| *Transfer* | | | | | | | | | | | |
| InfLoRA | – | $72.26^{\pm 0.56}$ | $36.19^{\pm 0.64}$ | $38.46^{\pm 0.39}$ | $55.22^{\pm 1.65}$ | $73.19^{\pm 0.55}$ | $39.32^{\pm 1.54}$ | $80.29^{\pm 0.91}$ | $51.19^{\pm 1.16}$ | $55.05^{\pm 0.51}$ | $55.69^{\pm 0.24}$ |
| CoDyRA | – | $74.3^{\pm 0.52}$ | $36.8^{\pm 0.23}$ | $44.2^{\pm 0.56}$ | $\mathbf{69.9^{\pm 0.56}}$ | $\mathbf{83.5^{\pm 0.23}}$ | $42.8^{\pm 0.18}$ | $\mathbf{88.9^{\pm 0.42}}$ | $64.6^{\pm 0.47}$ | $\mathbf{63.4^{\pm 0.56}}$ | $63.2^{\pm 0.28}$ |
| MoRA | – | $\mathbf{74.5^{\pm 0.51}}$ | $\mathbf{38.1^{\pm 0.24}}$ | $\mathbf{46.9^{\pm 0.56}}$ | $65.3^{\pm 0.44}$ | $82.9^{\pm 0.18}$ | $\mathbf{45.8^{\pm 0.31}}$ | $88.2^{\pm 0.15}$ | $\mathbf{65.1^{\pm 0.35}}$ | $62.9^{\pm 0.10}$ | $\mathbf{63.3^{\pm 0.26}}$ |
| *Average* | | | | | | | | | | | |
| InfLoRA | $20.49^{\pm 0.98}$ | $78.58^{\pm 1.02}$ | $48.5^{\pm 1.18}$ | $66.59^{\pm 1.51}$ | $71.83^{\pm 0.80}$ | $76.79^{\pm 0.34}$ | $61.45^{\pm 1.36}$ | $82.59^{\pm 0.86}$ | $55.3^{\pm 1.34}$ | $56.67^{\pm 0.59}$ | $62.48^{\pm 0.31}$ |
| CoDyRA | $41.4^{\pm 0.28}$ | $81^{\pm 0.38}$ | $58.7^{\pm 0.26}$ | $77.8^{\pm 0.47}$ | $83.4^{\pm 0.39}$ | $\mathbf{84.6^{\pm 0.28}}$ | $64.5^{\pm 0.14}$ | $\mathbf{90.4^{\pm 0.40}}$ | $67.2^{\pm 0.23}$ | $\mathbf{64.4^{\pm 0.47}}$ | $71.3^{\pm 0.18}$ |
| MoRA | $\mathbf{44.1^{\pm 0.24}}$ | $\mathbf{81.6^{\pm 0.34}}$ | $\mathbf{64.6^{\pm 0.34}}$ | $\mathbf{79.6^{\pm 0.37}}$ | $\mathbf{83.9^{\pm 0.36}}$ | $84.4^{\pm 0.15}$ | $\mathbf{66.5^{\pm 0.24}}$ | $89.7^{\pm 0.07}$ | $\mathbf{68.4^{\pm 0.38}}$ | $64.1^{\pm 0.09}$ | $\mathbf{72.7^{\pm 0.17}}$ |
| *Last* | | | | | | | | | | | |
| InfLoRA | $18.26^{\pm 0.49}$ | $\mathbf{82.36^{\pm 0.92}}$ | $46.57^{\pm 0.89}$ | $79.38^{\pm 2.22}$ | $76.16^{\pm 1.61}$ | $79.58^{\pm 0.60}$ | $95.74^{\pm 0.44}$ | $87.78^{\pm 0.85}$ | $71.11^{\pm 0.73}$ | $73.05^{\pm 0.19}$ | $70.99^{\pm 0.24}$ |
| CoDyRA | $\mathbf{37.7^{\pm 0.42}}$ | $81.5^{\pm 0.24}$ | $65.1^{\pm 0.63}$ | $89.9^{\pm 0.55}$ | $91.4^{\pm 0.38}$ | $85.5^{\pm 0.16}$ | $96.8^{\pm 0.08}$ | $\mathbf{93.3^{\pm 0.30}}$ | $77.3^{\pm 0.66}$ | $73.5^{\pm 0.21}$ | $79.2^{\pm 0.18}$ |
| MoRA | $\mathbf{37.7^{\pm 0.28}}$ | $81.5^{\pm 0.22}$ | $\mathbf{70.7^{\pm 0.49}}$ | $\mathbf{92.4^{\pm 0.20}}$ | $\mathbf{95^{\pm 0.34}}$ | $\mathbf{86^{\pm 0.13}}$ | $\mathbf{97.6^{\pm 0.19}}$ | $92.6^{\pm 0.10}$ | $\mathbf{81^{\pm 0.35}}$ | $\mathbf{74.7^{\pm 0.06}}$ | $\mathbf{80.9^{\pm 0.12}}$ |

## A.7 MORE COMPARISONS ON X-TAIL

To further validate MoRA's robustness, we evaluated it under other continual-learning task orderings, i.e., X-TAIL (Order 2), as shown in Table 17. The results align with those in Table 1, confirming that MoRA consistently achieves state-of-the-art performance.

## A.8 COMPUTATION COST

Table 10 summarizes the per-task trainable parameters of various continual-learning methods. Standard LoRA (Hu et al., 2021), CoDyRA (Lu et al., 2024), and O-LoRA (Wang et al., 2023a) each introduce $r(d_{\text{in}} + d_{\text{out}})$ new parameters per weight matrix. Mixture-of-Experts variants such as MoE-LoRA and MoE-Adapter (Yu et al., 2024) additionally train a router module to controll the usage of each LoRA experts, inducing $d_{\text{in}}$ additional paramters for each experts. LB-CL (Qiao & Mahdavi, 2024) introduce $r$ additional parameters, mimic the singular values of SVD.

By contrast, MoRA requires only $rd_{\text{in}} + kd_{\text{out}}$ activated trainable parameters per task, where $k \leq r$ is the activation budget, and the trainable parameter is at most the same as a standard LoRA. Despite the small number of parameter activated and trained, MoRA achieves superior continual learning performance, and reaches comparable performance in general fine-tuning with only one-third of the activated parameters of a standard LoRA.

**Trainable parameters and training GPU memory.** Beyond the estimated parameter counts in Table 10, in Table 18, we measured the actual trainable parameters for each continual-learning task and GPU memory usage, under the same settings as Table 1 in the main paper.

Table 17: Comparisons on X-TAIL (Order 2) for each domain in terms of "Transfer", "Average", and "Last" scores (%).

| Method | Cars | Aircraft | OxfordPet | Food | SUN397 | MNIST | Flowers | DTD | Caltech101 | EuroSAT | Average |
|---|---|---|---|---|---|---|---|---|---|---|---|
| *CLIP* | | | | | | | | | | | |
| Zero-shot | 66.1 | 23.5 | 86.7 | 84 | 63.7 | 46.7 | 63.6 | 37.3 | 76.8 | 36.7 | 58.5 |
| *Transfer* | | | | | | | | | | | |
| Zero-shot (Radford et al., 2021) | – | 23.5 | 86.7 | 84 | 63.7 | 46.7 | 63.6 | 37.3 | 76.8 | 36.7 | 57.7 |
| LwF (Li & Hoiem, 2017) | – | 20.0 | 74.1 | 79.6 | 58.1 | 34.1 | 48.9 | 27.7 | 64.4 | 15.1 | 46.9 |
| WiSE-FT (Wortsman et al., 2022) | – | 21.3 | 79.5 | 83.3 | 61.0 | 39.9 | 56.5 | 29.6 | 68.0 | 20.8 | 51.1 |
| ZSCL (Zheng et al., 2023) | – | 23.0 | 84.3 | 87.2 | 63.0 | 42.1 | 65.2 | 34.6 | 71.4 | 40.9 | 56.9 |
| MoE-Adapter (Yu et al., 2024) | – | 17.1 | 87.2 | 87.5 | 58.4 | 12.6 | 65.5 | 35.9 | 70.0 | 17.9 | 50.2 |
| RAIL-Primal (Xu et al., 2024) | – | 23.5 | 86.7 | 84 | 63.7 | 46.7 | 63.6 | 37.3 | 76.8 | 36.7 | 57.7 |
| CoDyRA (Lu et al., 2024) | – | 23.6 | 89.2 | 83 | 62 | 51 | 71.4 | 38 | 77.4 | 39 | 59.4 |
| MoRA | – | 23.6 | 88.7 | 83.4 | 62.6 | 51.2 | 69.9 | 39.3 | 77.5 | 39 | 59.5 |
| *Average* | | | | | | | | | | | |
| LwF (Li & Hoiem, 2017) | 49.0 | 27.4 | 69.7 | 83.0 | 65.7 | 42.2 | 63.5 | 33.1 | 68.5 | 17.5 | 52.0 |
| WiSE-FT (Wortsman et al., 2022) | 57.9 | 29.6 | 77.8 | 85.4 | 68.0 | 51.6 | 69.3 | 35.5 | 71.0 | 23.0 | 56.9 |
| ZSCL (Zheng et al., 2023) | 74.4 | 36.4 | 86.7 | 88.7 | 68.9 | 50.0 | 75.1 | 40.1 | 72.5 | 43.7 | 63.6 |
| MoE-Adapter (Yu et al., 2024) | 74.4 | 38.6 | 87.7 | 87.3 | 67.9 | 50.6 | 76.5 | 43.7 | 72.3 | 18.8 | 61.8 |
| RAIL-Primal (Xu et al., 2024) | 77.9 | 40.4 | 85.6 | 83.3 | 68.3 | 62.2 | 76.6 | 45.8 | 80.4 | 41.7 | 66.2 |
| CoDyRA (Lu et al., 2024) | 80 | 39.2 | 92.5 | 85.2 | 69.2 | 73.7 | 79.6 | 46.2 | 78.6 | 44.1 | 68.8 |
| MoRA | 80.2 | 40.1 | 92.5 | 84.7 | 70.1 | 74 | 80.1 | 48.7 | 78.4 | 44.4 | 69.3 |
| *Last* | | | | | | | | | | | |
| LwF (Li & Hoiem, 2017) | 29.6 | 17.5 | 63.0 | 83.8 | 67.7 | 44.9 | 79.3 | 44.8 | 84.6 | 39.0 | 55.4 |
| WiSE-FT (Wortsman et al., 2022) | 46.1 | 23.5 | 71.3 | 85.7 | 70.2 | 59.1 | 85.5 | 47.9 | 82.4 | 42.8 | 61.5 |
| ZSCL (Zheng et al., 2023) | 71.7 | 35.3 | 86.5 | 89.2 | 71.8 | 52.3 | 89.8 | 52.0 | 77.1 | 68.4 | 69.4 |
| MoE-Adapter (Yu et al., 2024) | 75.1 | 41.1 | 87.9 | 87.1 | 74.1 | 89.7 | 92.6 | 61.2 | 81.0 | 27.4 | 71.7 |
| RAIL-Primal (Xu et al., 2024) | 77.7 | 41.9 | 86.1 | 83.3 | 71.8 | 91.6 | 97.3 | 66.4 | 94.8 | 86.9 | 79.8 |
| CoDyRA (Lu et al., 2024) | 79 | 38.6 | 92.6 | 86.4 | 74.7 | 95.2 | 93 | 64.7 | 81.9 | 92.2 | 79.8 |
| MoRA | 79.3 | 38.9 | 93.1 | 85.4 | 74.9 | 96.4 | 94.1 | 69.9 | 82 | 92.9 | 80.7 |

Table 18: Trainable parameters and averaged training GPU memory per task.

| Method | Trainable Params. (Million) | GPU Mem. (MiB) |
|---|---|---|
| LWF (Li & Hoiem, 2017) | 129.6 | 32172 |
| ZSCL (Zheng et al., 2023) | 129.6 | 26290 |
| MoE-Adapters (Yu et al., 2024) | 59.8 | 22358 |
| CoDyRA (Lu et al., 2024) | 4.4 | 21770 |
| MoRA | 4.4 | 21090 |

LWF (Li & Hoiem, 2017) and ZSCL (Zheng et al., 2023) perform full-parameter fine-tuning, consuming the most parameters and memory. MoE-Adapters (Yu et al., 2024) maintains a router with 22 rank-64 adapter experts (top-2 activated) and a DDAS domain predictor. CoDyRA (Lu et al., 2024) trains a single rank-16 LoRA per task, reducing its footprint to 4.4 M parameters. MoRA introduces 16 rank-1 experts per task, with no additional router, for a total of 4.4 M trainable parameters and keeps a low gpu memory usage, thanks to our novel self-activated sparse mixture of ranks design.

## A.9 EXTENDED VISUALIZATIONS OF AGGREGATED RANK ACTIVATIONS

To complement the qualitative examples in Fig. 3, we include a statistical aggregation of rank-1 expert utilization over the entire test set for each task. This heatmap (Fig. 5) provides a global view of the routing behavior and confirms that the patterns observed in Fig. 3 are representative of the model's overall dynamics. The heatmap (Fig. 5) visualizes the activation ratio of each rank-1 expert (x-axis) across three scenarios (y-axis) in Fig. 3:

1. Task 1 Data (after Task 1): Consistent with the qualitative results in Fig. 3a, we observe statistically dominant usage of Rank 0 (airplane semantics) and Rank 11 (background/sky). Rank 11 shows higher overall activation frequency as it captures common background tokens, which constitute a larger portion of image patches than the object itself.

2. Task 1 Data (after Task 2): Crucially, the activation pattern remains virtually unchanged after training on Task 2. The heatmap shows near-zero activation for the newly introduced Task 2 experts (Ranks 16-31). This statistically proves that our routing mechanism is stable: "old" data does not drift to "new" experts, effectively preventing catastrophic forgetting.

3. Task 2 Data (after Task 2): We observe a distinct, dual-mode behavior:

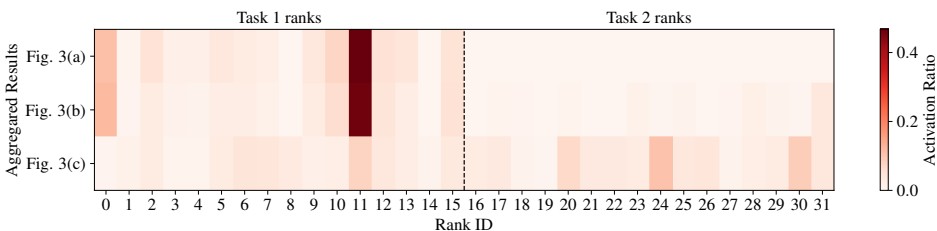

Figure 5: Averaged activation ratio of each rank-1 expert across three scenarios in Fig. 3

(a) Knowledge Reuse: Old Rank 11 is reactivated, confirming that the model reuses the generic "blue sky" feature for the new task.

(b) High Diversity: Unlike the concentrated pattern of Task 1 (Aircraft with homogeneous airplane images), Task 2 (Caltech101 with 101 diverse categories) utilizes a broad spectrum of new experts (e.g., Ranks 20, 24, 29, 30). This aligns with our design goal: fine-grained experts allow the model to dedicate different subspaces to the highly diverse semantics of the new task.

We explicitly chose to visualize activations at specific representative layers rather than averaging across the entire depth of the model. In the morden deep models, different layers specialize in distinct feature types (e.g., low-level textures vs. high-level semantics). Averaging expert usage across all layers would smooth out these distinct signatures and obscure the specialized routing behavior we aim to demonstrate.

### A.10 Extended Visualizations of Rank Activations

In Sec.4.4 (Fig.3), we illustrated rank activations during the learning of Task 1 and Task 2. Here, we extend these visualizations to additional tasks and scenarios in Fig.6 and Fig.7.

**MoRA retains task-specific semantics without forgetting.** Fig. 6 shows the activation maps for the same Task 1 image after training on Task 1 (a), Task 2 (b), and the last Task 10 (c). Patches corresponding to the airplane object are outlined in orange. In all three snapshots, Rank at index 0 remains consistently and exclusively activated for those airplane patches, demonstrating that MoRA has effectively memorized the airplane-specific knowledge into Rank 0. Even after 10 subsequent tasks, this pattern remains unchanged, indicating that later updates do not overwrite or interfere with the learned airplane representations. In other words, MoRA effectively memorizes and preserves task-relevant semantics, thereby mitigating catastrophic forgetting.

**MoRA encodes generic semantics that are reused across tasks.** Fig. 7 examines an input image from Task 9 before and after learning Task 9. Panel (a) shows the activation map of data from Task 1 after learning Task 1: Rank 11 (outlined in blue) already responds strongly to sky-background patches, demonstrating that MoRA has stored a generic "blue sky" concept in this rank. In panel (b), when we infer on the Task 9 image before training on Task 9, Rank 11 is again activated for the sky regions, confirming that MoRA reuses this shared knowledge for unseen data. Finally, panel (c) shows the activation map after learning Task 9: Rank 11 remains dedicated to the sky background, while newly initialized ranks specialize in the "car" object semantics. This persistent reuse of Rank 11 across tasks illustrates MoRA's ability to capture and retain common features as reusable memory slots, reducing redundancy and facilitating knowledge reuse.

### A.11 Statistical Analyses of Contributing Rank Activations

Fig. 8 and Fig. 9 plot the cumulative sum of averaged rank activations after training on all tasks, sorted in descending order, for several representative layers and locations within pre-trained models. The red dashed line marks the point at which 99% of the total activation mass is reached, allowing us to quantify how many ranks are truly contributing to the model's adaptation. Two key observations emerge:

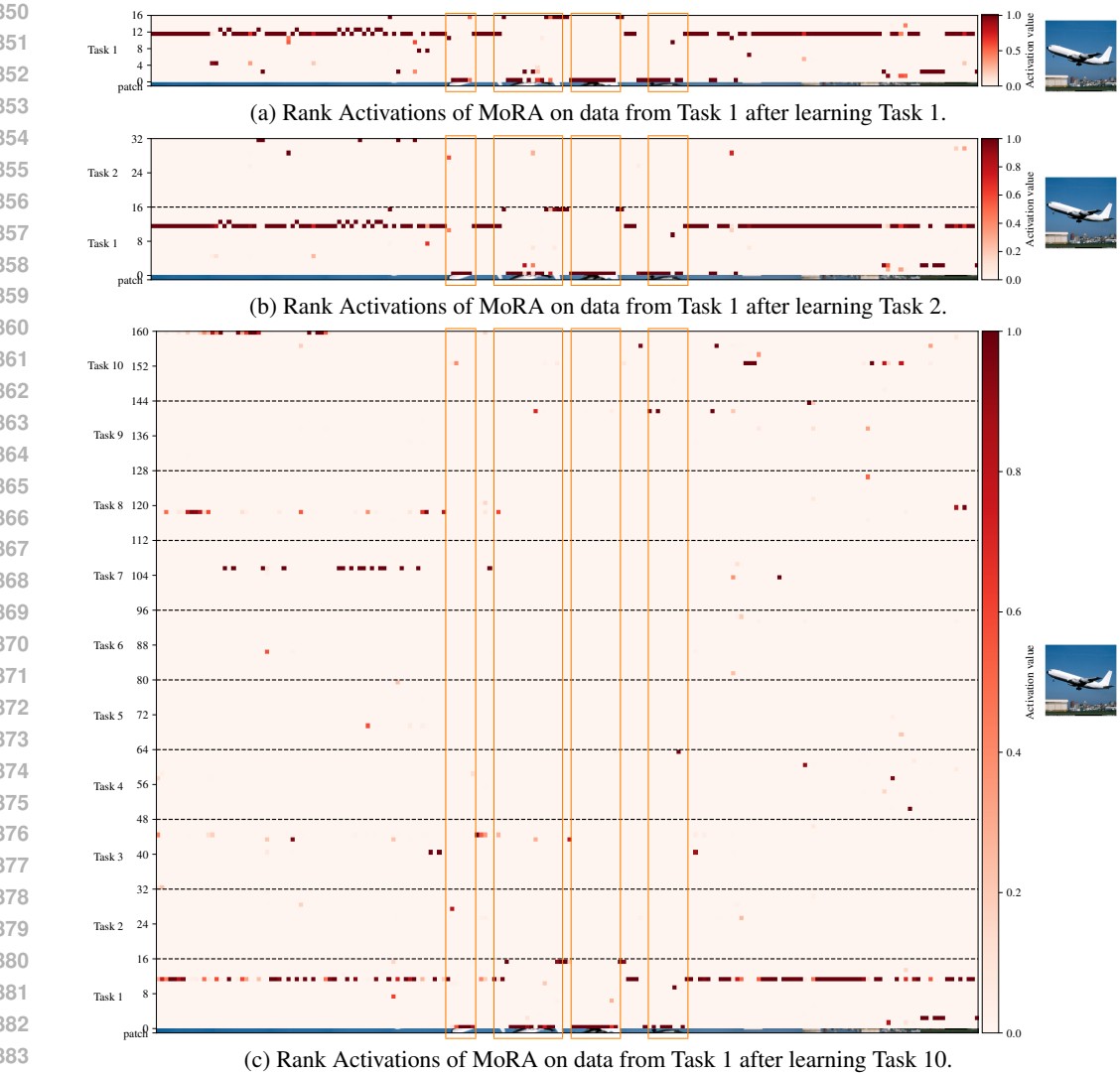

(a) Rank Activations of MoRA on data from Task 1 after learning Task 1.

(b) Rank Activations of MoRA on data from Task 1 after learning Task 2.

(c) Rank Activations of MoRA on data from Task 1 after learning Task 10.

Figure 6: Extended view of Fig. 3 illustrating **forgetting mitigation**. Regions corresponding to object semantics are highlighted with orange bounding boxes. Zoom in for details.

**1. Sparse Mixture: only a small subset of ranks is needed.** Across all layers and positions, we find that fewer than 10% of the total ranks suffice to capture 99% of the activations. This highlights the extreme sparsity of MoRA's self-activated mixture: most ranks remain dormant for any given input, while a compact set of highly relevant ranks drives the adaptation.

**2. Adaptive Activation: The number of ranks required varies by layers and modules.** The number of ranks needed to capture 99% of the cumulative activation mass varies across both layer depth and module type. For example, in Fig. 8, the MLP's output projection (`c-proj`) in Layer 1 of the vision encoder requires 16 ranks, whereas the same module in Layer 1 of the text encoder needs only 3 ranks.

To provide a broader view, Fig. 10 shows the required rank counts for every module in the pre-trained model. We observe that most attention modules requires around 6–12 ranks, while the second MLP projection generally demands more ranks in early layers, peaking in the first few blocks, and then steadily declines in deeper layers.

Coupling the rank activation budget with rank pruning, MoRA adapts the number of ranks needed to activate at each layer and module. This adaptive sparsity maximizes the efficient use of newly acquired knowledge during continual learning.

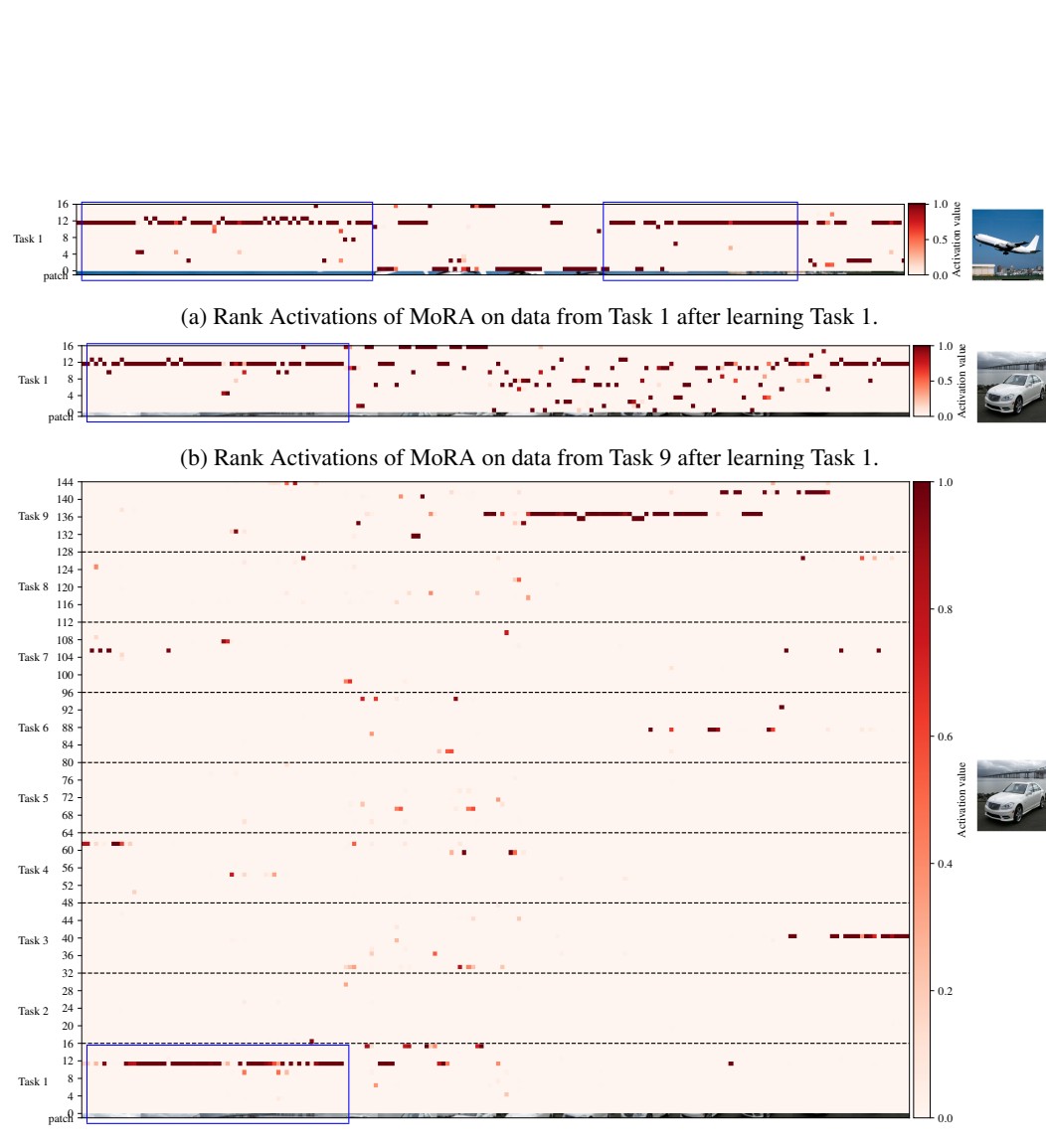

(a) Rank Activations of MoRA on data from Task 1 after learning Task 1.

(b) Rank Activations of MoRA on data from Task 9 after learning Task 1.

(c) Rank Activations of MoRA on data from Task 9 after learning Task 9.

Figure 7: Extended view of Fig. 3 illustrating **knowledge reuse**. Regions corresponding to generic input tokens (*e.g.* blue sky) are highlighted with blue bounding boxes. Zoom in for details.

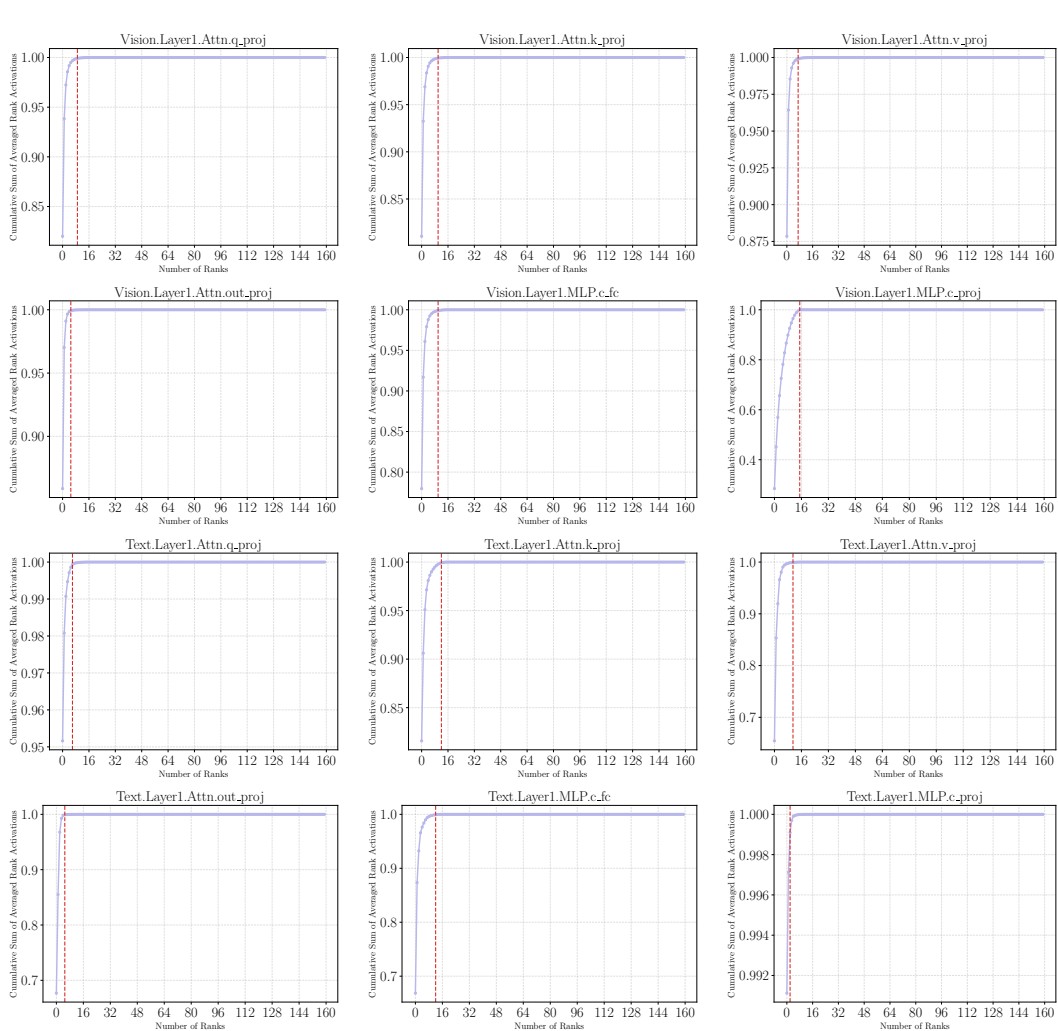

Figure 8: Statistical analyses on the number of ranks required to capture 99% of cumulative sum (indicated in red dashed line) of all rank activations. Activations were gathered from the model after training on all tasks, and results are shown for a representative selection of layers and positions within the pre-trained model.

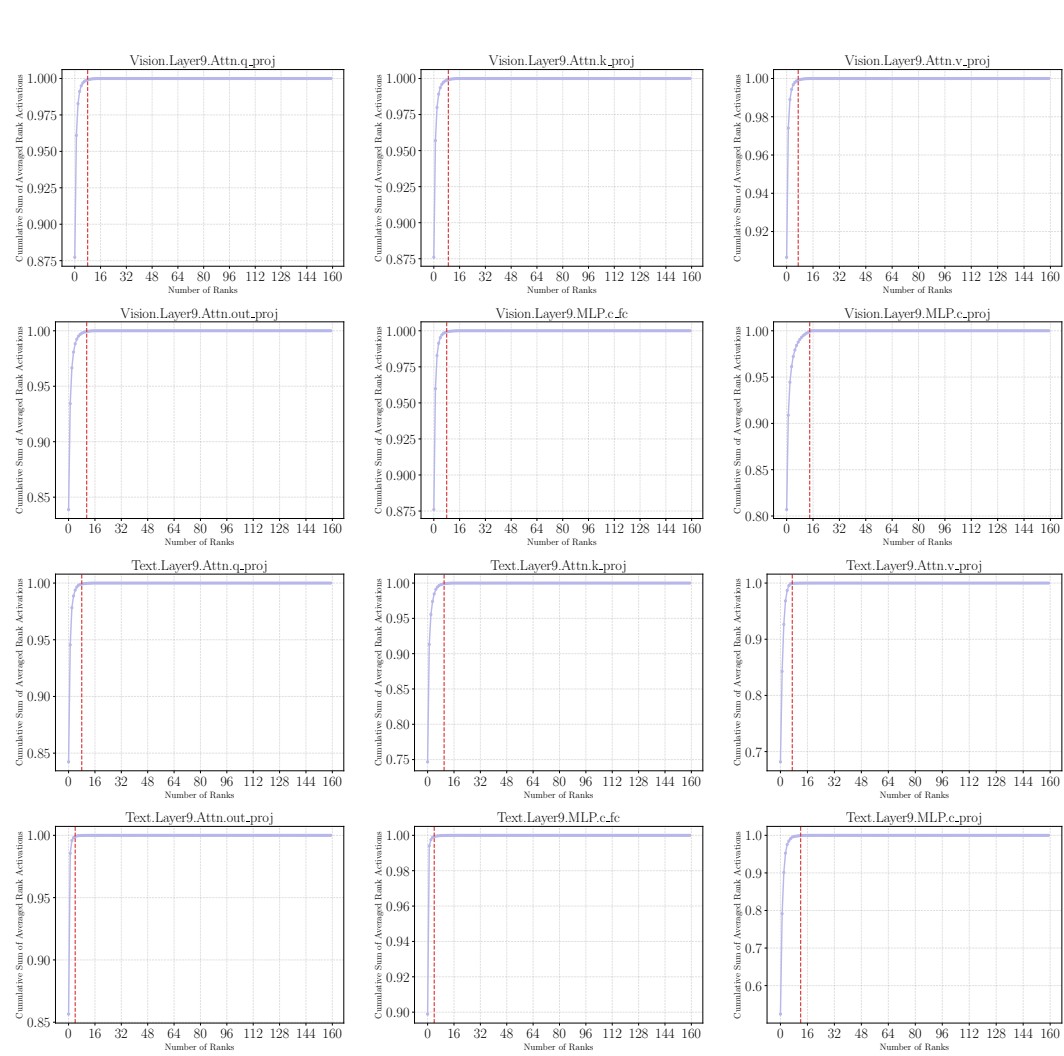

Figure 9: Statistical analyses on the number of ranks required to capture 99% of cumulative sum (indicated in red dashed line) of all rank activations. Activations were gathered from the model after training on all tasks, and results are shown for a representative selection of layers and positions within the pre-trained model.

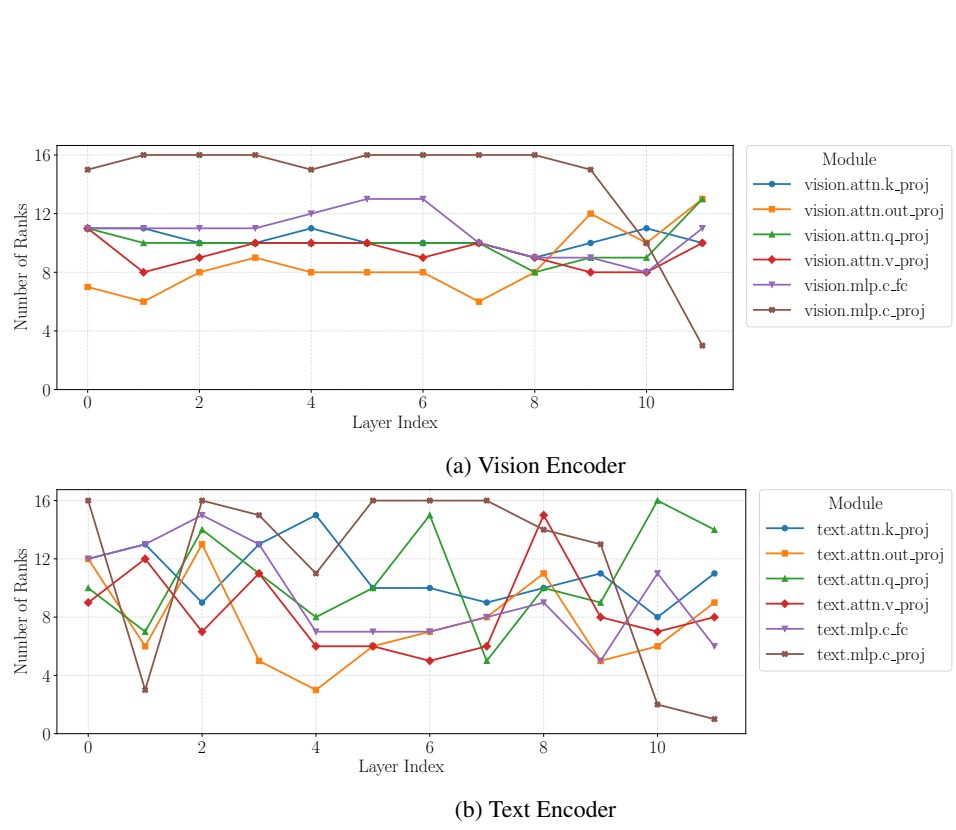

(a) Vision Encoder

(b) Text Encoder

Figure 10: Required ranks to capture 99 % of cumulative activations, shown across different pre-trained model layers and projection locations.

