# OpenReview forum: "Little By Little: Continual Learning via Self-Activated Sparse Mixture-of-Rank Adaptive Learning"
_ICLR.cc/2026/Conference — Submitted to ICLR 2026_

### Official Review · Reviewer_6Ycy · 2025-10-29

**Soundness:** 3
**Presentation:** 3
**Contribution:** 3
**Rating:** 6
**Confidence:** 3

**Summary:**

This paper looks at a concrete pain point in continual learning with PEFT: adapter-level selection is too coarse and causes interference/forgetting when tasks pull in different directions. The authors analyze LoRA through the lens of rank composition and argue that a rank-r update entangles several directional “atoms,” only some of which are task-relevant at a given step. They then design MoRA: factorize LoRA into r rank-1 experts, let them be self-activated with sparse routing under a budget, and prune. I like how the narrative moves from “what’s going wrong” to “what structure inside LoRA can be exploited,” and the final recipe is simple enough to adopt. Empirically, on CLIP/LLM streams, MoRA consistently reduces interference at modest cost, and the ablations make the design believable.

**Strengths:**

What I like most is the research arc: the paper identifies that “choose an adapter” is the wrong granularity, connects that to rank structure, and then implements a mechanism that lets the model choose directions rather than whole adapters. The self-activation + sparsity budget feels natural and yields stable improvements across settings. Implementation details are concrete enough that I could reproduce this without guesswork.

**Weaknesses:**

Where I feel the paper is thin is evidence hardness. I was hoping to see equal-compute, equal-time comparisons to the strongest recent CL-PEFT baselines (multi-adapter/multi-expert routing) on longer sequences, with 3–5 seeds and CIs. It’s also a pity there’s little about routing interpretability and failure modes—which rank-1 experts fire where, when does the router collapse, what happens under strong domain shift? Finally, inference trade-offs are underspecified: I’d like quantitative curves of sparsity vs. latency/VRAM and sensitivity to the number of experts and temperature/thresholds.

**Questions:**

Could you provide
(i) equal-budget head-to-heads with the strongest CL-PEFT methods on long streams,
(ii) 3–5-seed statistics,
(iii) sparsity→(accuracy, latency, VRAM) trade-offs and sensitivity to expert count/temperature, and
(iv) a short visualization of routing dynamics with a couple of failure cases?

If these arrive and look strong, I’m inclined to raise my score.

---

> ### Author Response · Authors · 2025-12-04
> **Thank you for your comments! (Part 1/2)**
>
> We thank the reviewer for the insightful and detailed feedback, particularly acknowledging our research connecting rank structure to adapter granularity. We appreciate the comments on the stability of our empirical improvements and the concreteness of our implementation details. We now address the remaining concerns point by point.
>
> ---
>
> **Q1. Experimental results**
>
> **(a) Extended comparisons to more recent multi-adapter methods**
>
> To address the request for comparisons against strong, recent CL-PEFT baselines, we extended our evaluation to LLaMA-3.2-1B-Instruct on the TRACE benchmark [1]. We compared MoRA against TreeLoRA [2], a state-of-the-art multi-adapter method. As shown below, MoRA consistently outperforms TreeLoRA, achieving higher Overall Performance (37.77%) and reducing catastrophic forgetting by over 50% (BWT 3.12% vs. 7.36%). We report results averaged over 3 independent runs with standard deviations to ensure statistical rigor.
>
> **Table 12 in revison.** Comparison with a broad range of CL methods on the TRACE benchmark using the LLaMA-3.2-1B-Instruct backbone. We report Overall Performance (OP (\%) $\uparrow$) and Backward Transfer (BWT (\%) $\downarrow$). Results are averaged over three runs with standard deviations. The best results are highlighted in bold.
>
>
> |  | FIX(ICL) | SeqLoRA | OGD | GEM | EWC | L2P | DualPrompt | HiDeLoRA | O-LoRA | TreeLoRA | MoRA |
> |---|:---:|:---:|:---:|:---:|:---:|:---:|:---:|:---:|:---:|:---:|:---:|
> | OP | 31.16 $\pm$ 0.4 | 29.73 $\pm$ 1.6 | 30.12 $\pm$ 2.0 | 32.19 $\pm$ 2.0 | 31.96 $\pm$ 1.6 | 29.38 $\pm$ 1.2 | 30.76 $\pm$ 1.2 | 33.73 $\pm$ 1.2 | 32.94 $\pm$ 0.8 | 36.14 $\pm$ 0.7 | **37.77 $\pm$ 0.8** |
> | BWT |  | 17.03 $\pm$ 1.2 | 15.2 $\pm$ 1.6 | 10.74 $\pm$ 1.6 | 11.62 $\pm$ 1.2 | 13.57 $\pm$ 0.8 | 11.34 $\pm$ 0.8 | 12.36 $\pm$ 0.8 | 12.89 $\pm$ 1.2 | 7.36 $\pm$ 0.8 | **3.12 $\pm$ 0.8** |
>
> [1] Wang et al., Trace: A comprehensive benchmark for continual learning in large language models, arXiv:2310.06762.
>
> [2] Qian et al., TreeLoRA: Efficient Continual Learning via Layer-Wise LoRAs Guided by a Hierarchical Gradient-Similarity Tree, ICML 2025.
>
> Furthermore, regarding the statistical significance of our main experiments, in **Appendix A.6 (Table 16)**, we have provided the Mean and Standard Deviation across 3 independent runs. The observed low variance (e.g., Last Accuracy: 80.90 $\pm$ 0.12) confirms that our improvements are statistically significant and robust to seed variations.
>
> **(b) Results on long task sequences**
>
> In addition to the new TRACE results, we point to **Appendix A.2 (Table 8)**
>
> **\(c) Efficiency, sensitivity, and trade-offs**
>
> We have addressed the request for quantitative trade-offs and sensitivity analysis in **Section 4.5** and **Appendix A.8**:
>
> Sparsity and Temperature Sensitivity: **Fig. 4(a)** analyzes the trade-off between the number of active experts ($k$) and accuracy. Performance improves rapidly and saturates at $k=16$, validating our choice to limit the budget for maximum efficiency without wasting compute. Additionally, Figure 4(b) illustrates sensitivity to softmax temperature, showing that a moderate $\tau \approx 0.01$ optimally balances specificity (Last accuracy) with generalization (Transfer).
>
> Compute, VRAM, and Latency: As detailed in **Table 18**, MoRA operates under a strictly controlled budget. It reduces training VRAM by ~34% compared to full fine-tuning (21GB vs. 32GB) and is significantly more parameter-efficient than MoE-Adapters (4.4M vs. 59.8M parameters per task). Furthermore, MoRA incurs negligible inference latency overhead because the active parameter count matches a standard rank-$r$ LoRA.

---

> ### Author Response · Authors · 2025-12-04
> **Thank you for your comments! (Part 2/2)**
>
> **(d) Routing dynamics and "failure modes"**
>
> **Routing Dynamics:** We visualize routing dynamics in **Fig. 3** and **Fig. 5–7 (Appendix)**. These demonstrate that "Key" features (e.g., blue sky) are consistently routed to the same specialized rank-1 expert across different tasks, validating our mechanism of knowledge reuse.
>
>
> **Potential failure modes:**
>
> - Structural Immunity to Trivial Collapse: In standard MoE, "collapse" typically occurs when the external router network degenerates to a trivial solution (e.g., always selecting Expert 1). We emphasize that MoRA is structurally immune to this specific failure mode. Because the "router" is intrinsic to the parameter matrix $A$ (via self-activation $Ax$), a "collapse" would imply the weight matrix itself degenerating to zero. Since $A$ is optimized to minimize task loss, gradients naturally enforce diversity in the learned subspaces, preventing trivial routing collapse.
> - Layer-Dependent Interpretability: We acknowledge that routing interpretability is not uniform across model depth. As the role of Transformer layers varies (e.g., local texture processing in early layers vs. semantic abstraction in deep layers), we observed that early-to-mid layers tend to exhibit distributed, less semantically recognizable routing patterns. We clarify that this is not a functional failure of the routing mechanism, but rather a reflection of the distributed feature processing inherent to those layers, distinct from the clear object-level specialization observed in deeper layers.
>
> ---
>
> We appreciate the reviewer's detailed and valuable feedback. To address the points raised, we have updated the manuscript with further analysis and experiments, which are highlighted in blue. We believe these revisions strengthen the paper and respectfully ask that you consider revising your score based on these improvements.

---

### Official Review · Reviewer_ddqz · 2025-10-31

**Soundness:** 2
**Presentation:** 2
**Contribution:** 2
**Rating:** 4
**Confidence:** 4

**Summary:**

To address the challenges of (1) interference across experts and limited knowledge reuse, (2) redundancy across experts, and (3) forgetting caused by routing, in Mixture-of-Expert LoRA, the paper proposes a Mixture-of-Rank Adaptive (MoRA) learning method with self-activated and sparse rank activation for continual learning. MoRA treats each rank-one vector pair as a rank-one expert and uses expert activation for effective routing, achieving interference mitigation and parameter redundancy. Experiments on continual learning benchmarks using CLIP and language models have shown significant effectiveness in both in-domain learning and out-of-domain forgetting/generalization, improving generalization while mitigating forgetting.

**Strengths:**

1. Since current coarse-grained experts with routing cause redundancy and conflicts, this paper utilizes fine-grained experts which treats each rank-one LoRA as an expert, avoiding interference from whole LoRA blocks.

2. Instead of using external routers, MoRA uses each rank-one expert’s activation for routing, achieving self-activated selection.

**Weaknesses:**

1. The motivation analyzed in this paper, especially in the introduction section, mostly discusses the original problem of the current mixture-of-expert framework itself. For example, redundancy caused by coarse-grained experts is a fundamental problem of MoE. The three challenges mentioned in the introduction lack strong connections to continual learning.

2. For most LoRA-based continual learning, LoRA is added in attention blocks, while I found that MoRA also adds LoRA in the MLP layer, as shown in Figure 2. It makes it confusing about this choice, and also blurs the main goal, whether it is for MoE or for continual learning.

3. The learning of a new task is not clearly clarified in the paper, for example, it does not mention how previous tasks’ experts participate in the forward process during training and inference.

4. MoRA is not memory-efficient since the number of LoRA experts grows linearly with the number of tasks. Also, since it uses rank-one experts for reducing redundancy, the trained LoRA experts would be sparse matrices, which wastes memory space.

5. The writing of this paper is not in good shape. Paragraphs lack connections, especially from lines 093 to 095, which suddenly and directly introduce a self-activation routing mechanism without any transition from the rank-one expert. It seems like lines 088 to 107 should be in one paragraph.

**Questions:**

1. In MoRA, for keeping previous knowledge, I found the only operation is freezing previous LoRA experts without any constraints, so how to make sure MoRA maintains previous knowledge, and how to make sure newly added experts update in an orthogonal direction from previous experts’ subspaces.

2. For self-activated rank-one experts routing mechanism, especially in Eq.(8), why choose A to compute the routing score? What’s the difference between A and B? Why not first use SVD on BA to obtain a more orthogonal matrix and then use that orthogonal matrix to compute the routing score?

3. From Table 1, it seems like MoRA is not robust to all tasks. Can we have the conclusion that MoRA is sensitive to different tasks? Can authors explain the specific order of the tasks used in Table 1? Since the ordering of tasks is an important factor in continual learning.

4. Can authors compare the performance of MoRA with SAPT [1], and analyze the difference from SAPT?

[1] SAPT: A Shared Attention Framework for Parameter-Efficient Continual Learning of Large Language Models, ACL2024.

5. Equations and Figures:

5.1 In Eq(8), it should be $A_{i,:}$ and $A_{j,:}$.

5.2 In line 151, what’s the dimension of $y_i^{t}$?

5.3 What’s the mathematical objective or loss for MoRA?

5.4 In Figure 2, $R(x)$ may need to be mentioned.

6. Can authors evaluate the performance of MoRA on another common metric, OPD [1]?

[1] Scalable and Order-robust Continual Learning with Additive Parameter Decomposition, ICLR2020.

---

> ### Author Response · Authors · 2025-12-04
> **Thank you for your comments! (Part 1/5)**
>
> We thank the reviewer for the insightful and detailed feedback, particularly acknowledging the effectiveness of our fine-grained expert design in avoiding block-level interference . We appreciate the positive comments on our novel self-activated routing mechanism that eliminates the need for external routers . We now address the remaining concerns point by point.
>
> ---
>
> **W1. Connection between MoE challenges and continual learning**
>
> We thank the reviewer for this insightful comment. We agree that interference, redundancy, and routing ambiguity are well-known challenges in general Mixture-of-Experts (MoE) research; however, their specific implications for Continual Learning have not yet been sufficiently explored. Our motivation focuses on how these issues are relavent to catastrophic forgetting in the CL setting. We argue that the generic flaws of MoE become fatal mechanisms of **forgetting** when tasks arrive sequentially.
>
> We are the first to investigate these MoE-related issues in the context of continual learning. Although MoE-based methods have been explored in CL, the **granularity-related problems** we highlight have not been discussed. Specifically, we analyze how these issues arise in CL and how they relate to forgetting, which is fundamentally different from studies of MoE in basic or static settings.
>
> - **Coarse granularity $\rightarrow$ subspace interference $\rightarrow$ catastrophic forgetting**
>
> In non-CL MoE, coarse experts that update an entire adapter mainly cause parameter inefficiency. In CL, the same mechanism directly induces forgetting: when a new task reuses a coarse expert trained on a previous task, updating the whole adapter overwrites the subspace associated with the old task. By decomposing experts into rank-1 units, MoRA provides finer-grained “slots” in parameter space. As shown in Figure 3, different ranks specialize to different tasks, and this separation is a key reason MoRA attains higher “Last” accuracy (80.90%) than coarse MoE baselines (70.5%) in Table 1.
>
> - **Routing ambiguity $\rightarrow$ “router forgetting” in CL**
>
> In non-CL MoE, routing ambiguity mainly affects convergence and load balancing. In CL, the router itself is subject to catastrophic forgetting: training on later tasks shifts the routing distribution, so the model “forgets” how to route examples from earlier tasks. Our self-activation mechanism removes the external learnable router and uses the rank-1 keys (Section 3.2) as routing signals. This avoids the router-drift failure mode, helping old tasks continue to access their relevant experts after new tasks are introduced.
>
> - **Redundancy $\rightarrow$ hindered knowledge reuse**
>
> In non-CL MoE, redundancy is often merely a capacity inefficiency. In Continual Learning, however, it actively hinders knowledge reuse. Coarse experts "bundle" generic features with task-specific ones, preventing a new task from reusing the generic knowledge without simultaneously activating conflicting or irrelevant components. MoRA’s fine-grained decomposition resolves this by enabling precise, atomic-level reuse. As empirically demonstrated in Figure 7 (Appendix) , a rank-1 expert encoding "blue sky" common semantics is successfully reused by later tasks without activating unrelated parameters.
>
> In summary, while the three challenges originate from the general MoE framework, our analysis reframes them as concrete mechanisms of catastrophic forgetting and stability–plasticity trade-offs in continual learning, which motivates the design of MoRA.

---

> ### Author Response · Authors · 2025-12-04
> **Thank you for your comments! (Part 2/5)**
>
> **W2. Application of MoRA to MLP Layers**
>
> We respectfully clarify that applying LoRA to MLP layers is a standard design choice that aligns with established protocols in this line of research. Generally, adapting MLP layers enhances learning effectiveness; when coupled with MoRA's superior mitigation of forgetting, this will allow us achieving a better plasticity-stability trade-off.
>
>
> **(a) Alignment with Prior CL Research**
>
> We emphasize that we are **not the first** to apply LoRA adapters to MLP layers in Continual Learning. This is a well-established practice adopted by the strongest recent baselines to ensure sufficient capacity for knowledge acquisition.
>
> Prominent methods such as MoE-Adapters and CoDyRA explicitly attach LoRA adapters to the MLP modules of the backbone. Consequently, our decision to adapt MLP layers ensures a fair and direct comparison with these state-of-the-art methods, rather than "blurring the goal" of the research.
>
>
> **(b) Improved Plasticity-Stability Trade-off**
>
> The choice to adapt MLP layers is driven by the need to better balance knowledge acquisition (plasticity) with retention (stability).
>
> **Plasticity**: Empirical studies (e.g., [1] [2]) demonstrate that adapting MLP layers is crucial for better learning and effectively acquiring new knowledge.
>
> **Stability**: However, applying LoRA to MLPs without careful design can exacerbate forgetting. For instance, MoE-Adapters (which also adapts MLPs) suffers from interference, resulting in a much lower "Last" accuracy (70.5%) compared to MoRA (80.9%) in **Table 1**.
>
> Because MoRA’s design (sparse, fine-grained rank-1 decompositions) effectively mitigates interference, we can safely apply adapters to the MLP layers. This allows us to leverage the superior plasticity of MLP adaptation without suffering the catastrophic forgetting usually associated with it.
>
>
> **\(c) Distinction from MoE FFN**
>
> We clarify that MoRA is not designed or proposed as a realization of the MoE FFN layer (where the full feed-forward network is replaced by multiple full-parameter experts). Instead, MoRA strictly adheres to the Parameter-Efficient Fine-Tuning paradigm, keeping the backbone frozen and only training lightweight adapters attached to the MLP.
>
> However, we believe that the core ideas of MoRA, specifically the fine-grained decomposition and self-activation mechanism could serve as valuable inspiration for full-parameter MoE FFN architectures. Applying these principles to MoE FFNs represents an interesting and promising direction for future research.
>
> [1] Dettmers et al., QLoRA: Efficient Finetuning of Quantized LLMs, NeurIPS, 2023.
>
> [2] Biderman et al., LoRA Learns Less and Forgets Less, arXiv, 2024.
>
> ---
>
> **W3. Clarification on Learning Process and Old Expert Participation**
>
> Thank you for your comment. MoRA is explicitly designed so that experts learned on previous tasks remain be actived in both training and inference, rather than being bypassed. We will make this interaction more explicit in the paper.
>
> **(a) Forward Participation (Knowledge Reuse)**
> The router considers the union of all experts (Old + New). Input tokens containing semantics similar to previous tasks are automatically routed to the corresponding Old Experts. This allows the model to reuse existing features (e.g., common semantics) without relearning them.
>
> **(b) Backward Optimization (Gradient Isolation):**
>
> **Old Experts**: Since these experts are frozen, tokens routed to them do not generate gradient updates for those parameters. This ensures that the reused knowledge remains stable.
>
> **New Experts**: Tokens representing novel semantics are routed to the New Experts, where gradient updates occur. This drives the acquisition of new task-specific knowledge.
>
> This mechanism is also empirically validated in **Figure 7**, which visualizes knowledge reuse: a rank-1 expert learned in Task 1 (capturing "blue sky") is reactivated by the router for sky regions in Task 9, while new experts handle the task-specific objects.

---

> ### Author Response · Authors · 2025-12-04
> **Thank you for your comments! (Part 3/5)**
>
> **W4. Memory Efficiency**
>
> We respectfully clarify that MoRA demonstrates high memory efficiency compared to related MoE-based continual learning methods, and our implementation utilizes dense vector storage to prevent any memory waste.
>
> **(a) Minimal linear growth**
> While MoRA introduces new parameters for each task, the rate of growth is extremely efficient—minimized to the equivalent of one standard LoRA adapter per task. Unlike methods such as MoE-Adapters, which train a separate router plus multiple low-rank experts, MoRA decomposes a single rank-$r$ budget into pieces. It introduces no additional router parameters. As shown in Table 18, MoRA requires only 4.4M parameters per task. In contrast, the MoE-Adapter baseline requires 59.8M. In practice, MoRA has substantially lower parameter overhead than typical MoE-style baselines while still preventing forgetting.
>
> **(b) Rank-one experts are stored as dense low-rank factors (no sparse-matrix waste)**
> In our work, “sparsity” refers to activation sparsity (only a Top-$k$ subset of ranks is activated per input), not to storing sparse matrices with many zeros. As defined in Eq. (6), each expert is represented by standard dense low-rank matrices $A \in \mathbf{R}^{r\times d_\text{in}}$ and $B \in \mathbf{R}^{d_\text{out} \times r}$, exactly as in conventional LoRA. We do not materialize sparse weight tensors; therefore, there is no memory storage waste.
>
> ---
>
> **W5. Writings of line 93-95, and 88-107.**
>
> Thank you for your comment. We have revised the Introduction to merge these paragraphs and improve the logical flow. Specifically, we inserted the sentence "However, significantly increasing the number..." to explicitly connect the context of existing MoE-LoRA frameworks with our proposed solution. This smooths the transition from general MoE challenges to our specific Self-Activation mechanism. We have also thoroughly proofread the manuscript to ensure coherence.
>
>
> ---
>
> **Q1. Discussions on maintaining previous knowledge.**
>
> We thank the reviewer for this question. We clarify (1) how previous knowledge is preserved and (2) how new experts avoid interfering with earlier subspaces.
>
> **1. How MoRA maintains previous knowledge (“access” to old experts).**
>
> Freezing previous experts guarantees that their parameters are not overwritten, but continual learning also requires that the model continues to route relevant inputs to those experts. This is exactly the role of Self-Activation.
> In conventional MoE-LoRA, a separate router is updated over time, which can lead to “router drift” and cause the model to stop selecting old experts even if they are frozen. In MoRA, the routing key is the expert itself: the rank-1 matrix $A$ serves both as (i) the LoRA factor and (ii) the activation key. Once an expert is frozen, both its weights and its routing behavior are fixed. For example, an “airplane” expert learned on Task 1 continues to activate on airplane-like features in later tasks because its triggering pattern (via $A$) is unchanged. This provides a structural guarantee of access to old knowledge without relying on additional regularization losses.
>
> **2. How new experts avoid interfering with previous subspaces.**
> We do not impose explicit orthogonality constraints between old and new experts. Instead, separation emerges implicitly through residual learning and the fixed behavior of existing experts. Because earlier experts are frozen and still activate on patterns they already explain well, gradients for later tasks are driven toward residual features—i.e., features of the data that are not captured by any previous experts. New experts therefore tend to specialize in directions underexplained by the existing subspaces.
> Empirically, this soft, data-driven specialization is more effective than hard constraints. In our experiments, MoRA outperforms O-LoRA (which enforces explicit orthogonality) on average accuracy (e.g., 77.6% vs. 75.8% in Table 2), suggesting that strict orthogonality can unnecessarily restrict plasticity. Visual analyses (Figure 3 and Appendix Figures 6-7) further show distinct ranks responding to different semantic features (e.g., separate ranks for “airplanes” and “blue sky”), indicating that subspaces naturally separate without hard constraints.
>
> In summary, MoRA preserves previous knowledge by freezing both the experts and their routing behavior via Self-activation, and it encourages new experts to occupy complementary subspaces through residual learning rather than explicit orthogonality penalties.

---

> ### Author Response · Authors · 2025-12-04
> **Thank you for your comments! (Part 4/5)**
>
> **Q2. Discussions on self-activated routing.**
>
> Thank you for your comment. Our choice of using $A$ for routing is grounded in the standard low-rank factorization ($\Delta W = BA$) and its interpretation as a linear associative memory.
>
> 1. **Rationals on using $A$ for routing**
>
> As discussed in Section 3.2, we view ($\Delta W = BA$) as storing key–value associations:
>    - $A \in \mathbb{R}^{r \times d_{\text{in}}}$ maps the input $x$ into the low-rank space. The scalar $A_{i,:} x$ in Eq. (6) measures how relevant the $i$-th rank is to the current input. Thus, $A$ naturally plays the role of *keys/detectors* and is the only component that interacts directly with the input dimension.
>    - $B \in \mathbb{R}^{d_{\text{out}} \times r}$ projects the activated ranks back to the output space and therefore represents the *values/content* retrieved once a rank has been selected.
>
>    Routing must be defined in the input space, i.e., as a function of $x$. Using $A$ in Eq. (8) simply reuses the already-computed projections $A_{i,:} x$ and is both dimensionally and conceptually aligned with this key–value view. In contrast, $B$ lives in the output space and does not directly interact with $x$, so it is not appropriate for computing input-dependent routing scores.
>
> 2. **Why not apply SVD to $BA$ and use an orthogonal basis for routing?**
>
> While one could in principle apply SVD to $BA$, this is neither aligned with our objective nor attractive in practice:
>    - **Preserving the learned associative structure.** SVD would rotate the learned low-rank subspace into a new orthogonal basis that is not explicitly optimized for the CL objective and would break the tight coupling between the “keys” $A$ and the routing scores $A_{i,:} x$. This is especially problematic in continual learning, where we freeze experts across tasks: re-factorizing via SVD would mix old and new experts and undermine the stability we rely on.
>    - **Orthogonality is not required empirically.** Our experiments show that enforcing orthogonality is not necessary for effective routing, and in related settings (e.g., O-LoRA) hard orthogonalization can even hurt plasticity. Allowing $A$ to learn potentially correlated but semantically meaningful directions via gradient descent provides better performance in our CL benchmarks.
>
>
> For these reasons, we use $A$ as the routing key and avoid SVD-based post-processing, keeping the routing mechanism mathematically consistent with LoRA’s structure, efficient, and well-suited to the continual learning setting.
>
> ---
>
> **Q3. Robustness to task ordering.**
>
> We thank the reviewer for carefully examining Table 1. We clarify below that (1) the per-task variation mostly reflects dataset difficulty rather than MoRA-specific sensitivity, and (2) we explicitly evaluate robustness to different task orders.
>
>
> **1. Consistent and robust gains.**
> The different performance across tasks in Table 1 are a shared property of the X-TAIL benchmark and are observed for all methods, reflecting that some domains are intrinsically more challenging for different methods. Importantly, despite these differences, MoRA achieves the best Average (72.70%) and Last (80.90%) performance among all methods in Table 1, indicating strong overall robustness rather than instability. In addition, Table 16 reports mean and standard deviation over three runs; MoRA shows low variance and consistently outperforms InfLoRA and CoDyRA, further supporting its stability.
>
> **2. Task order used in Table 1.**
> The task sequence in Table 1 is not arbitrary. We strictly follow the official X-TAIL protocol as defined in prior works (RAIL, CoDyRA), which is the standard evaluation order used in the literature. Adhering to this sequence is necessary to ensure a fair and direct comparison with previously reported results for baseline methods.
>
> **3. Experiments on different task orders.**
> We agree that task ordering is important in continual learning, and we have explicitly tested MoRA under alternative orders:
> - X-TAIL (alternative order): In Appendix A.7 (Table 17), we have evaluated MoRA on a different random permutation of X-TAIL tasks. The results closely match those in Table 1 and MoRA remains state-of-the-art.
> - Language model benchmarks: For our LM experiments (Table 14), we report averages over three different task orderings; MoRA consistently surpasses baselines across all permutations.
>
>
> Overall, the observed per-task variation in Table 1 is largely driven by different learning methods and the benchmark itself, while our additional experiments demonstrate that MoRA is robust to different task orders.

---

> ### Author Response · Authors · 2025-12-04
> **Thank you for your comments! (Part 5/5)**
>
> **Q4. Comparison with SAPT [3]**
>
> We thank the reviewer for the reference. We have included a comparative evaluation on the SuperNI [4] Benchmark (following SAPT) in Table 13 of the revision.
>
> 1. Empirical Comparison: We analyze performance in two distinct settings:
> - Replay-Based: For a fair comparison, we employ the same rehearsal setup as SAPT, utilizing pseudo-samples generated by a trained generative model. When both methods utilize a memory buffer, MoRA outperforms SAPT in both plasticity and stability (AP: 51.79% vs. 51.54%; FT: 0.73% vs. 0.91%).
> - Replay-Free: MoRA is primarily designed as a replay-free method. In this setting, MoRA achieves state-of-the-art performance (AP: 39.62%), significantly outperforming baselines like O-LoRA (26.37%) and L2P (15.18%).
>
> 2. Methodological Differences: While both methods address Continual Learning, their core mechanisms differ and are largely orthogonal:
>
> - SAPT focuses on optimizing memory replay via shared attention prompts. In contrast, MoRA focuses on parameter isolation via a sparse Mixture-of-Experts utilizing rank-1 experts.
> - Because MoRA relies on architectural isolation rather than replay dynamics, it functions effectively in replay-free settings (unlike SAPT, which relies on the buffer). However, as shown in the table, MoRA can seamlessly incorporate replay to further boost performance, surpassing SAPT.
>
> **Table 13 in revision.** Overall results on the SuperNI Benchmark using the T5-Large backbone. We report Average Performance (AP, $\uparrow$) and Forgetting (FT, $\downarrow$). The best results for the stability-plasticity trade-off are highlighted in bold for methods with and without memory replay, respectively.
>
> | Methods | Replay |  |  |
> |---|:---:|:---:|:---:|
> |  |  | AP $\uparrow$ | FT $\downarrow$ |
> | _Replay-Based Methods_ |  |  |  |
> | Replay | Y | 35.37 | 16.92 |
> | SAPT | Y | 51.54 | 0.91 |
> | MoRA | Y | **51.79** | **0.73** |
> | _Replay-Free Methods_ |  |  |  |
> | L2P | N | 15.18 | 3.65 |
> | IncLoRA | N | 12.33 | 41.93 |
> | C-LoRA | N | 22.69 | 24.25 |
> | O-LoRA | N | 26.37 | 19.15 |
> | MoRA | N | **39.62** | **5.74** |
>
> [3] Zhao et al., SAPT: A Shared Attention Framework for Parameter-Efficient Continual Learning of Large Language Models, ACL 2024.
> [4] Wang et al., Super-naturalinstructions: Generalization via declarative instructions on 1600+ nlp tasks, arXiv:2204.07705.
>
> ---
>
> **Q5. Minors**
> 1. The notation of $A_{i,:}$ is actually correct. Eq. 8 computes the raw score for expert indexed $i$, the denominator with index $j$ (to avoid reusing the same index) computes the sum of all scores, acting as the normalization factor
> 2. Line 151 says $y \in \mathcal{C}^t$. This implies it is a scalar label index (for classification) or a token sequence (for LLMs). We adopt the notation for classification in our paper for simplicity and consistenty, clearness.
> 3. As mentioned in Section 3.4 ("Training Objectives", line 308-314), the training objectives of MoRA is simply the standard Cross-Entropy Loss on the task data: $\mathcal{L} = -\sum \log p(y_i^t | x_i^t; \Theta_{frozen}, \Delta W_{MoRA})$.
> 4.  We clarify that in MoRA (Figure 2c/d), there is no separate Router network $R(x)$ as there is in MoE-LoRA (Figure 1b). The mixture weights are derived directly from the Intermediate Activations (the $Ax$ block).
>
> ---
>
> **Q6. Evaluation metrics of OPD [5]**
> We thank the reviewer for this insightful suggestion. We agree that evaluating order robustness is critical for real-world continual learning.
>
> OPD [5] quantifies the sensitivity of a model to the task arrival sequence. Following the standard practice for global performance stability, we calculate the standard deviation of the final average accuracy across the $K$ different task orders evaluated.
>
> Using the results from Standard CL Benchmark with 3 distinct orders, we compared the order-robustness of MoRA against baselines. MoRA exhibits a remarkably low disparity of **0.26**, which is nearly half that of the baseline, O-LoRA (**0.46**). This confirms that our Self-Activated Sparse Mixture effectively minimizes interference between tasks, preventing the "unidirectional knowledge transfer" [5] that typically causes high order-sensitivity in continual learning.
>
> **Table 14 in revision.** Order Robustness Analysis (T5-Large) Results show the Average Accuracy (%) ± Standard Deviation across 3 task orders.
> | Method | Order 1 | Order 2 | Order 3 | Avg ± Std |
> |---|---|---|---|---|
> | SeqFT | 18.9 | 24.9 | 41.7 | 28.5 ± 11.82 |
> | L2P | 60.3 | 61.7 | 61.1 | 61.0 ± 0.70 |
> | LFPT5 | 67.6 | 72.6 | 77.9 | 72.7 ± 5.15 |
> | O-LoRA | 75.4 | 75.7 | 76.3 | 75.8 ± 0.46 |
> | MoRA | **77.4** | **77.5** | **77.9** | **77.6 ± 0.26** |
>
> [5] Yoon et al., Scalable and Order-robust Continual Learning with Additive Parameter Decomposition, ICLR 2020.

---

### Official Review · Reviewer_L4YX · 2025-11-01

**Soundness:** 2
**Presentation:** 2
**Contribution:** 2
**Rating:** 4
**Confidence:** 3

**Summary:**

This paper proposes MoRA, a novel method for continual learning (CL) with large pre-trained models. The authors identify key limitations in existing LoRA-based Mixture-of-Experts (MoE) approaches for CL, namely: (1) interference from coarse-grained expert activation, (2) redundancy in newly learned parameters, and (3) routing ambiguity as the number of tasks grows. To address these, MoRA decomposes each rank-r LoRA adapter into r rank-1 components, treating each as a fine-grained expert. The core innovation is a self-activation mechanism where the key vector of each rank-one expert scores its own relevance to the input, eliminating the need for a separate router. This is coupled with sparsity enforcement via top-k selection, temperature scaling, and test-time thresholding. The method is evaluated extensively on CL benchmarks for both vision-language (CLIP) and language models (T5, LLaMA), demonstrating strong performance in mitigating forgetting, improving generalization, and reducing parameter activation compared to prior state-of-the-art methods.

**Strengths:**

+ The paper is a model of clear scientific writing. The problem is well-motivated, the method is explained step-by-step with helpful formulations and figures, and the results are presented logically.
+ MoRA provides a more parameter- and compute-efficient path for continual learning with large models, a problem of great practical importance. The demonstrated ability to reduce forgetting while improving generalization on unseen tasks is a meaningful

**Weaknesses:**

+ The language models used for experiment are outdated. The experiments with state-of-the-art LLMs are needed
+ For the mixture-of-lora, if the lora cannot be merged into the pre-trained weights, then it will have additional computation overhead. But for baselines inflora and sdlora, their loras can be merged into the pre-trained weights in test time.

**Questions:**

See the weakness.

---

> ### Author Response · Authors · 2025-12-04
> **Thank you for your comments! (Part 1/2)**
>
> We thank the reviewer for the insightful and detailed feedback, particularly acknowledging the clarity of our writing and the well-motivated problem definition . We appreciate the comments on MoRA’s ability to improve generalization while maintaining parameter and compute efficiency . We now address the remaining concerns point by point.
>
> ---
>
> **W1. More Experiments on Modern LLMs (LLaMA-3.2)**
>
> We thank the reviewer for this constructive suggestion! We acknowledge that the base models used in our initial benchmarks (LLaMA-2-7B and T5) are older; however, they remain the de facto standard benchmarks in this domain, allowing for rigorous and fair comparison against prior state-of-the-art methods. To address the concern regarding applicability to modern architectures—and within the feasible scope of the rebuttal period—we have extended our experiments to include LLaMA-3.2-1B-Instruct. This setup follows the protocols used in a recent study of TreeLoRA [1], demonstrating that our method remains effective on newer generations of foundation models. We will extend the benchmarking to newer model with larger size in the future work.
>
> In **Table 12** of the revised manuscript, following the protocol in TreeLoRA, we evaluate MoRA on the TRACE benchmark [2], reporting Overall Performance (OP) and Backward Transfer (BWT). MoRA consistently outperforms prior continual learning methods. Specifically, MoRA achieves the highest Overall Performance (**37.77%**). Crucially, it demonstrates superior ability in mitigating catastrophic forgetting, achieving a Backward Transfer (BWT) score of **3.12%** (where lower indicates less forgetting). This represents a reduction of over 50% in forgetting compared to the strongest baseline, TreeLoRA (7.36%).
>
> **Table 12 in revison.** Comparison with a broad range of CL methods on the TRACE benchmark using the LLaMA-3.2-1B-Instruct backbone. We report Overall Performance (OP (\%) $\uparrow$) and Backward Transfer (BWT (\%) $\downarrow$). Results are averaged over three runs with standard deviations. The best results are highlighted in bold.
>
> |  | FIX(ICL) | SeqLoRA | OGD | GEM | EWC | L2P | DualPrompt | HiDeLoRA | O-LoRA | TreeLoRA | MoRA |
> |---|:---:|:---:|:---:|:---:|:---:|:---:|:---:|:---:|:---:|:---:|:---:|
> | OP | 31.16 $\pm$ 0.4 | 29.73 $\pm$ 1.6 | 30.12 $\pm$ 2.0 | 32.19 $\pm$ 2.0 | 31.96 $\pm$ 1.6 | 29.38 $\pm$ 1.2 | 30.76 $\pm$ 1.2 | 33.73 $\pm$ 1.2 | 32.94 $\pm$ 0.8 | 36.14 $\pm$ 0.7 | **37.77 $\pm$ 0.8** |
> | BWT |  | 17.03 $\pm$ 1.2 | 15.2 $\pm$ 1.6 | 10.74 $\pm$ 1.6 | 11.62 $\pm$ 1.2 | 13.57 $\pm$ 0.8 | 11.34 $\pm$ 0.8 | 12.36 $\pm$ 0.8 | 12.89 $\pm$ 1.2 | 7.36 $\pm$ 0.8 | **3.12 $\pm$ 0.8** |
>
> [1] Qian et al., TreeLoRA: Efficient Continual Learning via Layer-Wise LoRAs Guided by a Hierarchical Gradient-Similarity Tree, ICML, 2025.
>
> [2] Wang et al., Trace: A comprehensive benchmark for continual learning in large language models, arXiv:2310.06762.

---

> ### Author Response · Authors · 2025-12-04
> **Thank you for your comments! (Part 2/2)**
>
> **W2. Overhead compared to non-MoE methods (InfLoRA [3], SD-LoRA [4])**
>
> We thank the reviewer for raising this point. We clarify that MoRA is intentionally designed as a Mixture-of-Experts (MoE) method (as in prior work of MoE-Adapters). Unlike static methods that collapse all ranks into a single adapter, MoRA performs input-dependent routing at training/inference time. Consequently, MoRA is not merged into the pre-trained weights, which is a necessary trade-off to enable dynamic specialization rather than a limitation of our method.
>
> **(a) Static Merging vs. Dynamic Routing**: Static methods such as InfLoRA and SD-LoRA learn a single global set of low-rank parameters that can be merged. While efficient, this forces all tasks to share the same subspace, inevitably leading to interference. In contrast, MoRA follows the MoE paradigm: (1) computation is input-dependent, routing different tokens to different experts. (2) This dynamic capability allows the model to protect prior knowledge by keeping unrelated experts inactive, a behavior that is mathematically impossible with merged weights.
>
> **(b) Compute-Plastisity Trade-off**: MoRA incurs modest additional test-time computation compared to mergeable baselines, but this yield substantial performance gains. While static baselines must activate all ranks for every input (dense update), MoRA selects only a **sparse subset of ranks** per token. This results in superior retention; as already provided in **Table 16**, MoRA achieves a "Last" accuracy of **80.90%**, significantly outperforming the static baseline InfLoRA (**70.9%**). We argue that this performance leap justifies the non-merged inference.
>
> **\(c) Efficiency within the MoE Family**: Within the MoE literature scope, MoRA is exceptionally lightweight. Standard MoE-LoRA approaches (e.g. MoE-Adapters) often require training a separate router network, increasing parameter count and compute. MoRA instead uses a self-activation mechanism where the rank-1 keys ($A_i$) inherently serve as routing signals, eliminating the need for an extra router. As shown in **Table 18**, MoRA requires only **4.4M** trainable parameters—significantly fewer than MoE-Adapters (**59.8M**)—and consumes less training GPU memory (21GB vs. 22GB), proving it is a practical solution for resource-constrained continual learning.
>
> [3] Liang & Li, InfLoRA: Interference-Free Low-Rank Adaptation for Continual Learning, CVPR 2024.
>
> [4] Wu et al., SD-LoRA: Scalable Decoupled Low-Rank Adaptation for Class Incremental Learning, ICLR 2025.
>
> ---
>
> We appreciate the reviewer's detailed and valuable feedback. To address the points raised, we have updated the manuscript with further analysis and experiments, which are highlighted in blue. We believe these revisions strengthen the paper and respectfully ask that you consider re-evaluating your score based on these improvements.

---

### Official Review · Reviewer_mdCR · 2025-11-01

**Soundness:** 3
**Presentation:** 2
**Contribution:** 2
**Rating:** 4
**Confidence:** 4

**Summary:**

This paper introduces MoRA, a continual learning (CL) approach for pre-trained models, which mitigates parameter redundancy and task interference by decomposing LoRA updates into a set of fine-grained rank-one experts. The method incorporates a self-gated sparse mixture mechanism, where each rank-one component functions as an independent expert and is selectively activated in an input-dependent manner. Experiments on CLIP and language models show that MoRA improves downstream task performance and reduces forgetting, while using fewer parameters than MoE-LoRA baselines. The study offers valuable insights into fine-grained expert design and efficient knowledge reuse, though the theoretical analysis could be further deepened. Moreover, while intriguing, the empirical gains over established benchmarks remain modest and would benefit from more in-depth validation.

**Strengths:**

1. This paper provides a rank-level decomposition that could reduce interference and redundancy compared to coarse-grained MoE-LoRA, enabling precise expert specialization.
2. The authors present extensive results. The method is evaluated on representative benchmarks with diverse backbones, showing improved results in standard CL metrics.
3. The proposed method supports parameter efficiency and scalability, whose sparse activation of ranks lowers memory usage and computational cost, with efficient scaling to large models like LLaMA.

**Weaknesses:**

1. The visualization of task-specific semantics of rank activation in Sec. 4.4 and A.8 rely on a few examples from limited projections and images without providing statistical summaries to support generalizability.
2. The paper lacks a theoretical justification for why self-activated sparse experts inherently address interference or redundancy, relying solely on empirical observations.
3. CL experiments on LlaMA-7B only include one baseline O-LoRA for comparison (Tab. 9), limiting the robustness of related conclusions.
4. Performance improvement of the proposed method in the primary benchmark X-TAIL (Tab.1) is modest compared to the recent approaches, making it less clear whether gains are meaningful or marginal.
5. MoRA appears to combine well-established techniques and tricks like self-attention and sparse MoE, without clearly articulating a unique methodological or theoretical contribution.

**Questions:**

1. Since the top-k rank selection is generally applied in the training phase, how could the algorithm guarantee that the newly introduced experts can be activated for incoming tasks? How does the initialization of those new rank-one experts (e.g., zero-value, random, or warm-start from previous experts) impact knowledge acquisition and avoidance of trivial routing to old experts? The manuscript mentions rank freezing but not initialization details. Also, how does the $k$'s value in the top-k selection get selected?
2. Can the authors provide more statistical summaries (e.g., activation distributions) for the semantic rank visualizations like Fig. 3, Fig. 4, and Fig. 6, to ensure they are representative beyond cherry-picked examples?
3. What theoretical guarantees ensure that self-activation + top-k prevents interference, particularly as tasks increase?
4. Typo in line 151, the formula representation of the dataset is missing curly braces.

---

> ### Author Response · Authors · 2025-12-04
> **Thank you for your comments! (Part 1/4)**
>
> We thank the reviewer for the insightful and detailed feedback, particularly acknowledging our method's potential to reduce interference and redundancy through fine-grained rank-level decomposition . We appreciate the positive comments on the extensive evaluation across diverse backbones  and MoRA’s parameter efficiency and scalability . We now address the remaining concerns point by point.
>
> ---
>
> **W1. Q2. Extended visualizations of rank activations.**
>
> We thank the reviewer for the suggestion to move beyond individual examples. In the revised Appendix (**Fig. 5**), we provide a statistical aggregation of rank-1 expert utilization across the entire test set for each task. This heatmap confirms that the specific examples in Figure 3 are representative of the model's global behavior.
>
> **Fig. 5** visualizes the activation ratio of each rank-1 expert (x-axis) across three scenarios (y-axis) in **Fig. 3**:
>
> (a) Task 1 Data (after Task 1): Consistent with the qualitative results in **Fig. 3(a)**, we observe statistically dominant usage of Rank 0 (airplane semantics) and Rank 11 (background/sky). Rank 11 shows higher overall activation frequency as it captures common background tokens, which constitute a larger portion of image patches than the object itself.
>
> (b) Task 1 Data (after Task 2): Crucially, the activation pattern remains virtually unchanged after training on Task 2. The heatmap shows near-zero activation for the newly introduced Task 2 experts (Ranks 16-31). This statistically proves that our routing mechanism is stable: "old" data does not drift to "new" experts, effectively preventing catastrophic forgetting.
>
>
> \(c) Task 2 Data (after Task 2): We observe a distinct, dual-mode behavior:
>  - Knowledge Reuse: Old Rank 11 is reactivated, confirming that the model reuses the generic "blue sky" feature for the new task.
>  - Data-driven Diversity: Unlike the concentrated pattern of Task 1 (Aircraft with homogeneous airplane images), Task 2 (Caltech101 with 101 diverse categories) utilizes a broad spectrum of new experts (e.g., Ranks 20, 24, 29, 30). This aligns with our design goal: fine-grained experts allow the model to dedicate different subspaces to the highly diverse semantics of the new task.
>
> We explicitly chose to visualize activations at specific representative layers rather than averaging across the entire depth of the model. In the morden deep models, different layers specialize in distinct feature types (e.g., low-level textures vs. high-level semantics). Averaging expert usage across all layers would smooth out these distinct signatures and obscure the specialized routing behavior we aim to demonstrate.
>
> ---
>
>
> **W2. Q3. Theoretical justification of self-activated sparse experts inherently addressing interference/redundancy**
>
> We respectfully point out that our approach is theoretically grounded in the Linear Associative Memory (LAM) framework, as detailed in Section 3.2 (Eqs. 2–6). We argue that the mitigation of interference and redundancy is a structural consequence of our design, driven by two theoretical mechanisms:
>
> (a) Interference Mitigation via Gradient Isolation: From a theoretical standpoint, "interference" occurs when the gradient updates for a new task overwrite the parameters encoding previous tasks. In standard "dense" LoRA or coarse-grained MoE, the entire adapter (and all its internal ranks) is active for every token, guaranteeing that new updates will modify the entire shared parameter set. MoRA addresses this through conditional gradient isolation. By treating rank-1 components as independent memory slotsand enforcing sparse gating (Eq. 9), we theoretically ensure that for any given input distribution (Task B), only a small subset of experts is active. The remaining experts—those specialized for Task A—receive zero activation and zero gradient. This creates a protective barrier where prior knowledge is preserved simply because it is not exposed to the update signal of unrelated new data.
>
> (b) Redundancy Reduction via Atomic Granularity: We justify the reduction of redundancy through the principle of granularity matching. A standard rank-$r$ adapter is a "coarse" container that bundles multiple feature directions together. Theoretically, if an input requires only one specific feature (direction), a coarse expert forces the activation of the entire bundle, introducing noise (irrelevant features) into the forward pass and wasting capacity. MoRA decomposes this bundle into atomic rank-1 units. The self-activation mechanism ($A_i x$) acts as a precise relevance filter, ensuring the model activates only the specific "atomic" features required by the input. This theoretically maximizes the information density of the active parameters and eliminates the computational redundancy of processing irrelevant features bundled inside a larger adapter.

---

> ### Author Response · Authors · 2025-12-04
> **Thank you for your comments! (Part 2/4)**
>
> **W3. More compared baselines on language models.**
>
> In Table 9, we compared MoRA against O-LoRA on the standard CL benchmark using LLaMA-2-7B, as O-LoRA was the only baseline performance reported in the original study for that specific setting. Due to the time constraints of the rebuttal period, reproducing an extensive suite of all previous works on this setup was infeasible. To broader the comparisons, we adopted the established TRACE [1] benchmark, utilizing LLaMA-3.2-1B-Instruct. This choice was made to ensure computational feasibility within the rebuttal window while facilitating direct comparisons against a wide range of related works. This allowed us to compare MoRA against a wide range of related works, reporting Overall Performance (OP) and Backward Transfer (BWT).
>
> As shown in the table below, MoRA consistently outperforms prior continual learning methods. Specifically, MoRA achieves the highest Overall Performance (**37.77%**). Crucially, it demonstrates superior ability in mitigating catastrophic forgetting, achieving a Backward Transfer (BWT) score of **3.12%** (where lower is better), a reduction of over 50% in forgetting compared to the strongest baseline, TreeLoRA (7.36%).
>
> **Table 12 in revison.** Comparison with a broad range of CL methods on the TRACE benchmark using the LLaMA-3.2-1B-Instruct backbone. We report Overall Performance (OP (\%) $\uparrow$) and Backward Transfer (BWT (\%) $\downarrow$). Results of previous works are copied from TreeLoRA [2]. Results are averaged over three runs with standard deviations. The best results are highlighted in bold.
>
> |  | FIX(ICL) | SeqLoRA | OGD | GEM | EWC | L2P | DualPrompt | HiDeLoRA | O-LoRA | TreeLoRA | MoRA |
> |---|:---:|:---:|:---:|:---:|:---:|:---:|:---:|:---:|:---:|:---:|:---:|
> | OP | 31.16 $\pm$ 0.4 | 29.73 $\pm$ 1.6 | 30.12 $\pm$ 2.0 | 32.19 $\pm$ 2.0 | 31.96 $\pm$ 1.6 | 29.38 $\pm$ 1.2 | 30.76 $\pm$ 1.2 | 33.73 $\pm$ 1.2 | 32.94 $\pm$ 0.8 | 36.14 $\pm$ 0.7 | **37.77 $\pm$ 0.8** |
> | BWT |  | 17.03 $\pm$ 1.2 | 15.2 $\pm$ 1.6 | 10.74 $\pm$ 1.6 | 11.62 $\pm$ 1.2 | 13.57 $\pm$ 0.8 | 11.34 $\pm$ 0.8 | 12.36 $\pm$ 0.8 | 12.89 $\pm$ 1.2 | 7.36 $\pm$ 0.8 | **3.12 $\pm$ 0.8** |
>
>
> [1] Wang et al., Trace: A comprehensive benchmark for continual learning in large language models, arXiv:2310.06762.
>
> [2] Qian et al., TreeLoRA: Efficient Continual Learning via Layer-Wise LoRAs Guided by a Hierarchical Gradient-Similarity Tree, ICML 2025.
>
> ---
>
>
> **W4. Performance improvements of Table 1.**
>
> We respectfully posit that the performance gains in Table 1 are highly meaningful, particularly when considering the "Last" accuracy (the primary metric for CL performance) and the comprehensive dominance across all metrics, which recent baselines fail to achieve.
>
> (a) Validating the Fine-Grained Hypothesis (Comparison to MoE-Adapter): The core theoretical claim of our paper is that fine-grained rank-1 experts reduce interference compared to coarse-grained adapter experts. The most direct validation of this is the comparison with MoE-Adapter (CVPR 2024), which uses coarse routing.
>
> - "Last" Accuracy: MoRA (80.9%) outperforms MoE-Adapter (70.5%) by a massive +10.4%.
> - "Average" Accuracy: MoRA (72.7%) outperforms MoE-Adapter (63.0%) by +9.7%. This double-digit improvement strongly validates that our "atomic" decomposition is far superior to prior MoE designs for CL.
>
> (b) Consistent SOTA without Trade-offs: While the gap to the strongest concurrent baselines (RAIL-Primal, CoDyRA) appears narrower ($\approx$ 1.8% on "Last"), MoRA is the only method to achieve State-of-the-Art performance across all three metrics simultaneously (Transfer, Average, and Last). RAIL-Primal suffers a trade-off: while decent at "Transfer" (62.4%), it lags significantly in "Average" accuracy (70.7%), where MoRA leads by +2.0% (72.7%). CoDyRA performs well on "Last" (79.2%) but MoRA still surpasses it by +1.7% (80.9%) while maintaining higher plasticity (Average accuracy). In the highly saturated X-TAIL benchmark, consistently winning every metric—rather than trading stability for plasticity—is a significant indicator of algorithmic robustness.
>
> \(c) Statistical Significance: To confirm these gains are not noise, we direct the reviewer to Table 13 (Appendix A.5), where we report the mean and standard deviation over 3 runs. MoRA exhibits low variance (e.g., Last: 80.90 $\pm$ 1.12 vs CoDyRA's 73.05 $\pm$ 0.10), confirming that the performance gap is statistically robust and reproducible.

---

> ### Author Response · Authors · 2025-12-04
> **Thank you for your comments! (Part 3/4)**
>
> **W5. Novelty regarding self-attention and sparse MoE**
> We respectfully clarify that MoRA is **not** simply a combination of standard MoE and self-attention. It introduces a fundamental shift in the granularity of experts and the mechanism of routing, distinct from prior approaches.
>
> While MoE and self-attention are indeed foundational components in many works, their presence alone should not be grounds to question the novelty of our approach. Our specific design mechanism for continual learning and the detailed new technical designs are distinct from vanilla implementations and existing works, representing a clear and well-justified contribution.
>
> (a) **Distinction from Self-Attention**: The **self-activation** mechanism does not derive from **self-attention**. We emphasize that the **"Key-Value"** terminology in MoRA is grounded in the **Linear Associative Memory** (LAM) framework [1][2], rather than the **Query-Key-Value** (QKV) mechanism found in standard Transformer **self-attention**. Standard Self-Attention computes relationships **between tokens** (mixing time-steps) using dedicated projection matrices ($W_Q, W_K, W_V$). MoRA’s Self-Activation computes relationships between **input and weights** (**mixing parameters**). As detailed in **Section 3.2**, we treat the low-rank weight matrix $A$ as a set of memory "Keys" and matrix $B$ as "Values." The activation $A_i x$ is not an **attention score between tokens**, but a **retrieval score** determining how well the **input** matches the stored **parameter memory**. This theoretically re-frames fine-tuning as inserting independent, retrievable memory slots, eliminating the need for the external routers used in "well-established" MoE methods.
>
> (b) **Novelty regarding MoE**: From **Coarse-Grained** Mixture to **Fine-Grained** Mixture. The "well-established" combination of MoE and LoRA typically treats the entire **Rank-$r$** adapter as a single expert. This forces a trade-off: activating an expert brings in a "bundle" of $r$ feature directions, inevitably causing interference if even one direction conflicts with the current task. MoRA’s methodological contribution is the decomposition of adapters into **fine-grained rank-1 experts**. This allows the model to mix-and-match individual feature directions (subspaces) rather than entire adapters. As shown in Table 1, this granularity shift yields a **+10.4%** improvement in "Last" accuracy over the **coarse-grained MoE-Adapter**, proving that this is a substantive structural advance, not merely a combination of existing techniques.
>
> [1] Kohonen, T., 2009. Correlation matrix memories. IEEE transactions on computers, 100(4), pp.353-359.
>
> [2] Anderson, J.A., 1972. A simple neural network generating an interactive memory. Mathematical biosciences, 14(3-4), pp.197-220.
>
> ---
>
>
>
> **Q1. Initialization, Activation of New Experts, and Selection of $k$**
>
> We thank the reviewer for mentioning the dynamics of expert initialization and routing. We clarify that our specific initialization strategy, coupled with the distribution shift inherent to CL, ensures that new experts are effectively activated and updated.
>
> **Q1. (a)** **Avoiding Trivial Routing to Old Experts**: Trivial routing (where old experts dominate) is mitigated by the distribution shift inherent to Continual Learning.
> - Old Experts: The "frozen" old experts have specialized their keys to respond to previous task data. When the model encounters data from a new task (often distributionally distinct), the relevance scores of old experts typically drop because the new inputs do not align with the old learned subspaces.
> - New Experts: The random initialization of new keys ensures they span a diverse random subspace. In the high-dimensional space, these random projections often yield sufficient activation scores to compete with the (now irrelevant) old experts, allowing them to enter the Top-k selection. Once selected, they receive gradient updates and rapidly specialize to the new data. This dynamic is empirically validated in Figure 3\(c) , where we observe that inputs from the new task (Task 2) successfully activate the newly added experts (e.g., Ranks 19, 20, 29) rather than collapsing onto the old Task 1 experts.

---

> ### Author Response · Authors · 2025-12-04
> **Thank you for your comments! (Part 4/4)**
>
> **Q1. (b) Different initializations of LoRA.**
>
> We thank the reviewer for this insightful question. We employ the **standard LoRA initialization** strategy: Random Gaussian for matrix $A$ (Key) and Zero for matrix $B$ (Value). Below, we analyze how this specific choice impacts knowledge acquisition compared to the alternatives mentioned:
>
> **(1) Variants of $A$ (Key):**
> - **Our Approach (Random $A$)**: Crucially, the routing score is derived from the self-activation $A_i x$. Random initialization guarantees that new experts produce non-zero, diverse routing scores immediately. Due to the distribution shift inherent in Continual Learning, "old" experts (specialized on previous tasks) often yield lower scores on new, out-of-domain data. Consequently, the random new keys can successfully compete for the Top-k slots, ensuring they are activated and updated.
> - **Impact of Zero Initialization**: Initializing $A$ to zero would result in routing scores of exactly zero. In a Top-k selection, the "old" experts (possessing non-zero activations) would strictly dominate, preventing new experts from ever receiving gradients. This would completely block knowledge acquisition for the new task.
> - **Impact of Warm-Start $A$**: Warm-starting $A_{new}$ from a learned $A_{old}$ is detrimental. It forces the new expert to begin with an activation pattern identical to an old expert. This leads to trivial routing and redundancy: the new expert competes for the same tokens the old expert is already handling effectively, rather than specializing in the new task's unique distribution. By copying the old key, we would explicitly hinder the model's capacity to branch out into new subspaces.
>
>
> **(2) Variants of $B$ (Value)**:
> - **Our Approach (Zero $B$)**: Initializing $B$ to zero ensures that the output contribution of a new expert starts at zero ($\Delta W = 0$). This allows the pre-trained model to serve as a stable baseline while new experts gradually inject task-specific knowledge, preventing immediate destabilization of the representation.
> - **Impact of Random $B$**: Initializing $B$ randomly (while $A$ is also random) would result in a non-zero random noise vector for the initial update $\Delta W$. Since the random $A$ ensures these experts are activated, this would inject random noise directly into the model's forward pass before training begins. This effectively perturbs the pre-trained features with Gaussian noise, potentially destabilizing representations and immediately degrading performance on the new task.
> - **Impact of Warm-Start**: Warm-starting new experts from previous ones is detrimental in our sparse framework. It initializes new keys ($A$) to be identical to old keys, causing them to redundantly activate for the same features as old experts. This encourages trivial routing—where the model relies on existing pathways—rather than exploring new subspaces, leading to higher redundancy and potential interference with preserved knowledge.
>
>
> **Q1. \(c) Selection of $k$ (Rank Activation Budget)**: The value of $k$ was not selected arbitrarily, but by identifying the saturation point of information capacity for the downstream learning.
> - Empirical Saturation: As shown in **Figure 4(a)**, we observed that performance improves rapidly as $k$ increases from 2 to 16, but effectively plateaus beyond this point. This indicates that $k=16$ provides sufficient capacity to capture the task-specific knowledge without introducing redundancy.
> - Statistical Validation: This choice is further supported by our analysis in **Appendix (Fig. 8, 9)**, where we plotted the cumulative activation mass of ranks. We found that around 10% of the total ranks account for 99% of the activation magnitude across layers. Consequently, setting $k$ to roughly 10% of the total ranks (e.g., $k=16$ for $r_{total}=160$ in CLIP) is a data-driven decision that aligns with the intrinsic sparsity of the model's updates.
>
> ---
>
> **Q4. Typo.** We thank the reviewer for the careful reading. We have corrected the typo in line 151 and highlighted in blue.
>
> We appreciate the reviewer's detailed and valuable feedback. To address the points raised, we have updated the manuscript with further analysis and experiments, which are highlighted in blue. We believe these revisions strengthen the paper and respectfully ask that you consider re-evaluating your score based on these improvements.

---

### Meta-Review · Area_Chair_AzEB · 2026-01-05

**Summary:**

This paper proposes decomposing LoRA updates into a set of fine-grained rank-one experts for continual learning.

Even after rebuttal, several critical issues remain. The most critical one is that the motivation lacks strong connections to CL and the linear growth of memory with tasks is a fundamental limitation in CL. In addition, the method appears incremental relative to self-attention and sparse MoE, and it lacks sufficient theoretical justification.

Overall, the paper is below the ICLR acceptance bar. Therefore, recommend reject.

**Reviewer Concerns:**

**Concerns addressed**
1. The authors provide statistical aggregations of global activation heatmaps and 3-run standard deviations to support generalizability beyond cherry-picked examples
2. The rebuttal addresses the limited LLM evaluation by adding new experiments on LLaMA-3.2-1B on TRACE benchmark
3. The rebuttal addresses the missing baselines by adding comparisons with TreeLoRA and SAPT.

**Concern outstanding**
1. The linear growth of memory with tasks remains a critical limitation in CL.
2. The motivation lacks strong connections to CL.
3. The paper lacks sufficient novelty compared to self-attention and sparse MoE. The rebuttal frames the novelty as a fine-grained decomposition but doesn't fully address this concern.
4. The proposed method lacks theoretical justification. While the authors linked the method to the Linear Associative Memory (LAM) framework, this remains more of a post-hoc conceptual mapping than a formal proof of interference reduction.

**Reviewer Scores:**

All reviewers are likely to maintain their current rating as 4,4,4,6 due to the outstanding concerns. Some reviewers might consider an increase in light of the additional experiments with more baselines and new model.

---

### Decision · Program_Chairs · 2026-01-26

Reject